# DIVE: Scaling Diversity in Agentic Task Synthesis for Generalizable Tool Use

**Aili Chen**[♠♣] **Chi Zhang**[♣] **Junteng Liu**[♡♣] **Jiangjie Chen**[◇] **Chengyu Du**[♠♣] **Yunji Li**[♣] **Ming Zhong**[♣]
**Qin Wang**[♣] **Zhengmao Zhu**[♣] **Jiayuan Song**[♣] **Ke Ji**[♣] **Junxian He**[♡] **Pengyu Zhao**[♣] **Yanghua Xiao**[♠†]

[♠]Fudan University   [♣]MiniMax   [♡]Hong Kong University of Science and Technology   [◇]Independent

alchen20@fudan.edu.cn   shawyh@fudan.edu.cn

https://sheep333c.github.io/DIVE/

## Abstract

Recent work synthesizes agentic tasks for post-training tool-using LLMs, yet robust generalization under shifts in tasks and toolsets remains an open challenge. We trace this brittleness to insufficient diversity in synthesized tasks. Scaling diversity is difficult because training requires tasks to remain executable and verifiable, while generalization demands coverage of diverse tool types, toolset combinations, and heterogeneous tool-use patterns. We propose **DIVE**, an evidence-driven recipe that inverts synthesis order, executing diverse, real-world tools first and reverse-deriving tasks strictly entailed by the resulting traces, thereby providing grounding by construction. DIVE scales structural diversity along two controllable axes, tool-pool coverage and per-task toolset variety, and an Evidence Collection–Task Derivation loop further induces rich multi-step tool-use patterns across 373 tools in five domains. Training Qwen3-8B on DIVE data (48k SFT + 3.2k RL) improves by **+22** average points across 9 OOD benchmarks and outperforms the strongest 8B baseline by **+68%**. Remarkably, controlled scaling analysis reveals that diversity scaling consistently outperforms quantity scaling for OOD generalization, even with 4× less data.

## 1. Introduction

Recent work on agentic post-training increasingly relies on **synthesized agentic tasks**, improving LLMs' ability to use general-purpose tools such as web search and code execution (Yao et al., 2024; Xu et al., 2024; Liu et al., 2025b; Froger et al., 2025). However, in practical deployments, these models often struggle with the **open-ended diversity**

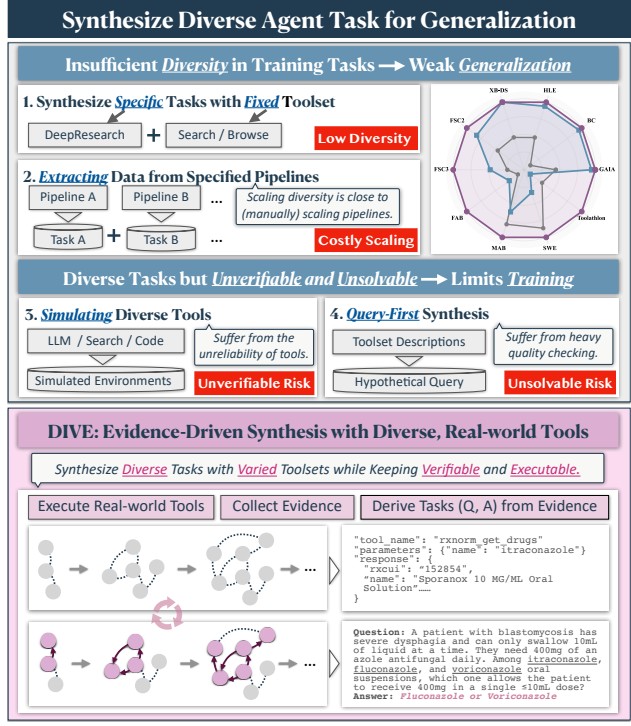

*Figure 1.* **Motivation and overview of DIVE. Top:** Fixed-toolset synthesis and pipeline pooling limit diversity and weaken generalization. **Middle:** Simulated tools and query-first synthesis for diverse tasks increase unverifiability/unsolvability risk, limiting agentic training. **Bottom:** DIVE performs evidence-first synthesis on diverse, real-world tools, producing verifiable and executable tasks. **Radar:** Gray: base model; Blue: trained on deep-research data synthesized with a fixed search/browse toolset (strong in-distribution but weak/negative transfer); Purple: trained on DIVE with matched data and training budget (robust generalization).

of tool use (Zhang et al., 2025): tasks range from open-domain queries (e.g., "what is the capital of Australia?") to domain-specific tasks such as clinical diagnosis, financial analysis, and software engineering (Jiang et al., 2025; Hu et al., 2025; Jimenez et al., 2023), while tools vary from general-purpose ones (e.g., web search) to specialized ones such as protein retrieval and email management (Mitchener et al., 2025; Xu et al., 2024). This motivates our primary

*Proceedings of the 43rd International Conference on Machine Learning*, Seoul, South Korea. PMLR 306, 2026. Copyright 2026 by the author(s).

†Corresponding author.

research question: *how can we improve the **generalization** of tool-using LLMs across real-world tasks and toolsets?*

We argue that a key bottleneck is **insufficient diversity in synthesized training tasks**, which limits generalization under task and toolset shifts. Most existing synthesis recipes scale data primarily by *quantity* or *difficulty*, but remain confined to **narrow** task families and **fixed** toolsets (Figure 1(1); e.g., deep-research tasks equipped with web search tools) (Liu et al., 2025b; Li et al., 2025d). As a result, while these agents perform well on in-distribution tasks, they often over-rely on rigid routines (e.g., search→browse loops) (He et al., 2025; Fu et al., 2025), leading to *poor generalization* or even *negative transfer* on new task families and toolsets (Figure 1 radar), e.g., when asked to perform clinical diagnosis using tools like `PatientLookup`.

However, **scaling diversity** while maintaining **data quality** is challenging because effective agentic training requires synthesized tasks to be both **verifiable** and **executable** for trajectory filtering and reward computation. This creates a fundamental tension: (1) *Structural Diversity*: beyond diverse tool types and per-task toolset combinations, tasks should involve heterogeneous multi-step tool-use patterns (e.g., *retrieval-only → retrieval-then-analyze*), rather than template substitutions (e.g., changing query entities); but (2) *Grounded Validity*: as diversity grows, ensuring every synthesized task remains solvable and verifiable under its specific toolset becomes increasingly difficult. Current approaches fail to reconcile this tension (Figure 1): *extracting* data from specialized pipelines is costly, as scaling diversity requires manually scaling pipelines (Liu et al., 2025a); *simulating* tool environments with LLMs or generic tools suffers from the unreliability of simulated tools, leading to *unverifiable risks* (Castellani et al., 2025; Mitra et al., 2024); and *query-first* synthesis on real tools suffers from heavy quality checking to mitigate the *unsolvable risk* of hypothetical queries (Qin et al., 2023; Shen et al., 2024).

To bridge this gap, we **invert the synthesis order** on **diverse, real-world tools**. Rather than generating task queries first and then checking validity post hoc, we execute tools first and derive tasks from the resulting traces. This yields **grounding by construction**: executability follows from real tool traces, and verifiability follows from observable tool outputs. Simultaneously, we scale structural diversity by expanding tool-pool coverage and per-task toolset variety; real executions yield tasks both grounded and structurally diverse, with heterogeneous tool-use patterns.

Specifically, we propose DIVE (Figure 1 bottom), an evidence-first recipe that automatically synthesizes **Di**verse, **V**erifiable, and **E**xecutable agentic tasks. Starting from *Retrieval* and *Processing* tool-use primitives, we construct three resource pools: 373 validated tools spanning general-purpose and four expert domains, domain-specific seed con-

cepts, and diverse query-only exemplars. Each synthesis cycle randomly samples a toolset, a seed, and exemplars, then runs a two-stage loop: (i) **Evidence Collection** interleaves multi-step reasoning with real tool use to gather logically related evidence and dynamically induce diverse tool-use patterns, and (ii) **Task Derivation** observes and reorganizes the accumulated evidence to reverse-derive grounded query–answer pairs strictly entailed by the traces; as evidence grows across iterations, it further refines tasks to remain grounded while increasing diversity. Finally, we apply these synthesized tasks to train agents via SFT and RL, validating their effectiveness for robust generalization.

In summary, our contributions are:

- We investigate diversity scaling in agentic task synthesis for generalizable tool use, identifying two coupled data requirements: *grounded validity* (verifiable/executable under assigned toolsets) and *structural diversity* (heterogeneous patterns beyond template variation).
- We propose DIVE, an evidence-driven recipe that synthesizes diverse, verifiable, and executable agentic tasks at scale by inverting the synthesis order: executing diverse, real-world tools first and reverse-deriving tasks.
- Extensive experiments demonstrate that DIVE significantly improves tool-use generalization. Our analysis reveals that diversity scaling outperforms quantity scaling, and that RL benefits are amplified by diverse training data.

**Conflict of Interest Disclosure.** Most co-authors are affiliated with MiniMax, whose large language model products may benefit from the agentic task synthesis methods studied in this paper for training tool-using agents.

## 2. Related Work

**Tool-Use Agents and Benchmarks.** Tool-use agents must operate under diverse constraints: varying toolsets, invocation protocols, and interaction environments (Qin et al., 2023; Tang et al., 2023; Yao et al., 2024). Benchmarks now reflect this diversity, spanning web research (Mialon et al., 2023; Wei et al., 2025; Chen et al., 2025), software engineering (Jimenez et al., 2023), domain applications (Hu et al., 2025; Choi et al., 2025; Jiang et al., 2025), and universal tool suites (Li et al., 2025b; Wang et al., 2025b; Guo et al., 2025). These benchmarks highlight a central challenge: *generalizable* tool use across shifting task distributions and toolsets, which is the focus of this work.

**Synthetic Data for Tool-Use Agent Training.** Scaling synthetic tasks and trajectories for SFT and RL is a prevailing paradigm for training tool-use agents (Qin et al., 2023; Tang et al., 2023; Mitra et al., 2024; Liu et al., 2025b; Li et al., 2025d;c). Most prior work designs synthesis pipelines tailored to fixed task types and toolsets to optimize agent performance, e.g., deep-research tasks equipped with general

web search tools (Liu et al., 2025b; Li et al., 2025d; Wu et al., 2025a;b; Qiao et al., 2025). However, this results in training data with limited diversity, hindering tool-use generalization to diverse unseen scenarios (He et al., 2025; Fu et al., 2025). A common engineering practice involves *extracting* data from specialized pipelines (Liu et al., 2025a; Team et al., 2025), but this heuristic is costly and scales poorly as each task type or environment demands a customized synthesis pipeline. To inherently scale diversity, other works attempt to *simulate* diverse toolsets via LLMs or generic tools (e.g., search, code execution) (Mitra et al., 2024; Fang et al., 2025; Castellani et al., 2025; Li et al., 2025e). While achieving scalability, they risk unstable mock execution where tasks solvable during synthesis may fail verification during training. Conversely, methods targeting *real* toolsets typically follow a *query-first* paradigm (Qin et al., 2023; Shen et al., 2024; Li et al., 2025b; Guo et al., 2025), creating a verification bottleneck: tasks derived from documentation are often non-executable, and manual verification is costly, hindering scalable RL. In contrast, DIVE guarantees executability and verifiability by construction via an *inverted* synthesis process on *diverse, real-world* tools.

## 3. DIVE

In this work, we aim to *improve tool-use generalization* by *scaling diversity in agentic task synthesis*. We propose DIVE, an automated recipe designed to achieve this goal while ensuring training stability. After introducing preliminaries (§3.1), we describe DIVE in three phases: (1) Diverse Synthesis Resource Preparation (§3.2), which builds decoupled pools of tools, seeds, and exemplars to support scalable synthesis; (2) Evidence-Driven Task Synthesis (§3.3), which reverse-derives tasks from grounded execution traces; and (3) Agentic Training with DIVE Tasks (§3.4), which optimizes the agentic LLM via supervised finetuning (SFT) and reinforcement learning (RL).

### 3.1. Problem Formulation

**Tool-Using Agent.** We formulate tool use as a sequential decision process. Given a task query $Q$ and a toolset $\mathcal{T}$, an agent policy $\pi_\theta$ performs interleaved reasoning and tool use (Yao et al., 2023) to solve the problem. At step $t$, the agent generates a thought $r_t$ and an action $a_t \in \mathcal{T}$ based on the history; the environment executes $a_t$ and returns an observation $o_t$. This yields a trajectory $\tau = (r_1, a_1, o_1, \ldots, r_T, a_T, o_T)$.

**Task Synthesis Objectives.** We aim to synthesize an agentic task dataset $\mathcal{D} = \{(Q^{(i)}, A^{(i)}, \mathcal{T}^{(i)})\}$, where each instance comprises a task query $Q^{(i)}$, a reference answer $A^{(i)}$ (for verification), and a unique toolset $\mathcal{T}^{(i)}$. To support **generalization** and effective **training**, $\mathcal{D}$ must satisfy four rigorous properties: (1) **Structurally Diverse**: Tasks should

cover diverse toolsets and exhibit heterogeneous tool-use patterns to support generalization. (2) **Verifiable**: Each task must have a deterministic verifier (e.g., by comparing the output to a reference answer) to ensure trajectory filtering and reward computation. (3) **Executable**: Each task must be solvable under its specific toolset $\mathcal{T}^{(i)}$, guaranteeing at least one feasible solution path to avoid optimization noise. (4) **Scalable**: The synthesis pipeline must be autonomous, enabling data volume to scale with compute resources.

### 3.2. Diverse Synthesis Resource Preparation

The diversity of synthesized data is inherently constrained by the richness of its underlying resources. Accordingly, prior to synthesis, we pre-construct a large, heterogeneous resource bank to support scalable and diverse task synthesis. We decompose this bank into three **diverse** and **decoupled** pools: (1) a **tool pool** defining a broad action space; (2) a **seed pool** providing long-tail semantic coverage; and (3) an **exemplar pool** offering heterogeneous structural priors. By decoupling these resources, we can exponentially expand task diversity through their independent sampling and recombination, covering a vast space of domain knowledge, tool capabilities, and reasoning structures.

**Tool Pool: Broad Action Space.** We start from two common generic tools: web search and code execution, which reveal a functional duality: search exemplifies the **Retrieval** primitive for acquiring external information, while code execution represents the **Processing** primitive for performing deterministic transformation. To systematically diversify tool use beyond generic utilities, we instantiate these two primitives with domain-specific tools across four expert domains: Finance, Biology, Medicine, and Academia. We construct the tool pool with a **Crawl–Validate** pipeline. (1) *Crawl.* Firstly, we crawl public APIs and wrap them for tool-calling, labeling each tool as Retrieval (e.g., `ncbi_search`) or Processing (e.g., `seq_translate`). (2) *Validate.* To ensure training stability, we filter candidates via unit tests for correctness, concurrency safety, and response consistency, yielding a final set of 373 robust tools (more details in Appendix B.1).

**Seed Pool: Diverse Semantic Anchors.** Synthetic generation can suffer from *topic collapse*, over-sampling generic, high-frequency concepts (Wang et al., 2023; Gudibande et al., 2024). To promote distributional diversity, we build a registry of **seed concepts** as anchors. We mine four domains: *Wikipedia* (Bridge, 2001), *PubMed* (National Library of Medicine (US), 2026), *NCBI* (Sayers et al., 2025), and *global stock exchanges* (Yahoo Finance, 2026), yielding $\sim$5,000 entity seeds per domain via LLM extraction. Anchoring synthesis on specific entities (e.g., *"Erlotinib"*) rather than generic terms (e.g., *"medicine"*) encourages broader exploration of sparse tool-space regions.

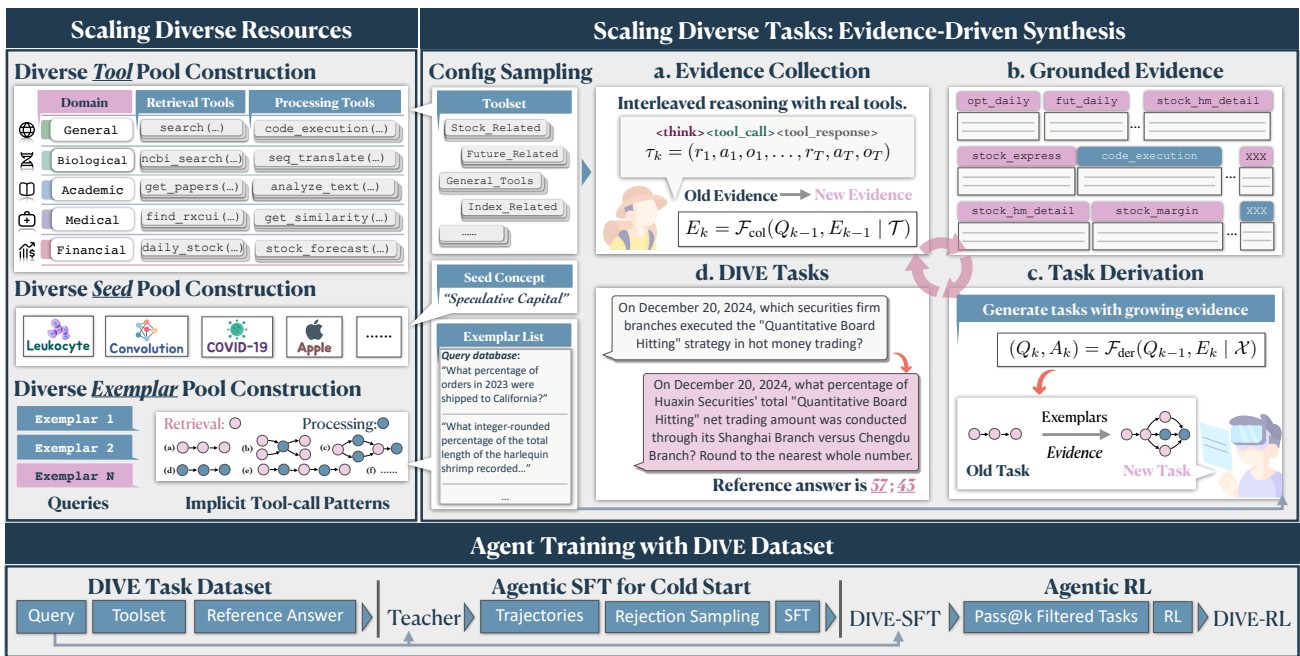

*Figure 2.* **Overview of the DIVE framework. (1) Diverse Synthesis Resource Preparation** (Left): We construct decoupled pools of tools (spanning general and expert domains), seed concepts, and query-only exemplars with implicit tool-use patterns. **(2) Evidence-Driven Task Synthesis** (Right): We randomly sample configurations and run an inverted loop where the model executes real tools to collect grounded evidence (a, b) and reverse-derives tasks (query-answer pairs) strictly entailed by traces (c, d), ensuring validity by construction. **(3) Agentic Training** (Bottom): The synthesized corpus supports effective SFT cold starts and RL using verifiable reference answers.

**Exemplar Pool: Heterogeneous Task Priors.** In contrast to synthesis methods constrained by fixed tasks and toolsets (Wang et al., 2025c; Li et al., 2025d; Wu et al., 2025a), open-ended generalization necessitates a broader spectrum of task forms. Accordingly, we construct a repository of query-only **exemplars** sourced from heterogeneous task families. Although exemplars contain no execution traces, each provides structural priors: (1) a *query phrasing*; and (2) an *implicit tool-use pattern*, e.g., *"Query database: what percentage of orders in 2023 were shipped to California?"* (Figure 2) implies a retrieve-then-compute structure. Drawing from diverse exemplars broadens the space of derived task forms (see Appendix B.2 for sources).

### 3.3. Evidence-Driven Task Synthesis

Given the diverse resource (§3.2), DIVE synthesizes training tasks via an **evidence collection and task derivation loop** (Figure 2 Right). Each synthesis samples a *synthesis configuration*, then *executes tools first* to collect grounded evidence (Figure 2a,b), and finally *derives* tasks that are strictly supported by the evidence (Figure 2c,d).

**Configuration Random Sampling.** To achieve diverse yet grounded synthesis, we begin each synthesis cycle by sampling a **synthesis configuration** $C = \{\mathcal{T}, S, \mathcal{X}\}$ (Figure 2 Right-Config Sampling). (1) *Seed Sampling:* We draw a seed concept $S$ from the seed pool to anchor the semantic context. (2) *Toolset Sampling:* Conditioned on the seed's do-

main, we sample a compatible toolset $\mathcal{T}$ ($|\mathcal{T}| \in [15, 50]$) to define the execution environment. (3) *Exemplar Sampling:* We sample a small set of query-only exemplars $\mathcal{X}$ (typically 3–5) as lightweight form-level cues. Randomly composing these components creates a vast space of configurations, providing diverse starting points for synthesis.

**Collect Evidence During Interleaved Reasoning with Real Tools.** At each iteration $k$, we invoke an *evidence collector* agent to expand the evidence frontier (i.e., grounded tool execution traces with outputs) by executing real tools under the configuration $\mathcal{T}$ (Figure 2a). The collector is conditioned on the evolving synthesis context: (i) the current inquiry $Q_{k-1}$ (initialized as $Q_0 = S$ and updated via the derivation step); (ii) the accumulated evidence $E_{k-1}$ (where $E_0 = \emptyset$); and (iii) the available toolset $\mathcal{T}$. Operating within these bounds, the agent performs a multi-step rollout (up to $T_{\max}$ steps) to produce a trajectory $\tau_k = (r_1, a_1, o_1, \ldots, r_T, a_T, o_T)$, where $r_t$ denotes the reasoning thought, $a_t$ the tool invocation (function and arguments), and $o_t$ the real execution return. We define this update as accumulating validated pairs $(a_t, o_t)$ from $\tau_k$ into the evidence set (Figure 2b):

$$E_k = \mathcal{F}_{\text{col}}(Q_{k-1}, E_{k-1} \mid \mathcal{T}). \tag{1}$$

By enforcing execution-first collection, we ensure that every evidence item in $E_k$ is grounded and replayable, imposing strict executability constraints on task derivation.

**Derive and Refine Tasks with Growing Evidence.** Following the collection step, we invoke a *task generator* LLM to synthesize a task grounded in the execution traces (Figure 2c,d). The generator maps the prior query state and current evidence to a new query–answer pair:

$$(Q_k, A_k) = \mathcal{F}_{\text{der}}(Q_{k-1}, E_k \mid \mathcal{X}), \qquad (2)$$

where $Q_0$ is initialized as the seed $S$ (and subsequent $Q_{k-1}$ are inherited). Conditioned on exemplars $\mathcal{X}$, the generator instantiates diverse query forms and implicit tool-use patterns (e.g., multi-hop retrieval or retrieve–compute pipelines) by composing evidence from $E_k$. Crucially, while $Q_k$ may vary in form, its content remains strictly grounded in $E_k$, and $A_k$ is derived directly from this evidence (see Appendix A for synthesized examples).

**Iterative Synthesis Loop.** We execute the collection–derivation loop for $K$ iterations, progressively increasing the diversity of both the evidence set and synthesized tasks (Appendix C) while keeping them grounded in real tool execution. Each iteration is confined to a sampled toolset $\mathcal{T}$: collection expands $E_k$ using only $\mathcal{T}$, and the derived query $Q_k$ becomes the basis for the next collection step, forming a closed-loop curriculum. This shared constraint guarantees executability and verifiability by construction: since $Q_k$ is instantiated by composing elements from $E_k$, its implicit solution corresponds to a sub-trace of tool calls under $\mathcal{T}$; therefore, a valid trajectory over $\mathcal{T}$ exists to recover the reference answer $A$; since $A$ is derived from tool-returned outputs, it is deterministically verifiable. We store refined tasks and aggregate the dataset for agentic training (§3.4):

$$\mathcal{D}_{\text{task}} = \{(Q_K^{(i)}, A_K^{(i)}, \mathcal{T}^{(i)})\}_{i=1}^N. \qquad (3)$$

### 3.4. Agentic Training with DIVE Tasks

We post-train an agent on the synthesized DIVE tasks with a two-stage scheme (Figure 2 Bottom): a supervised cold start to acquire reliable tool-calling, followed by reinforcement learning to improve robustness and generalization under diverse toolsets. Each training instance is a tuple $(Q, A, \mathcal{T})$, where $\mathcal{T}$ is the *same* toolset under which the task was synthesized, and $A$ is the reference answer.

**Agentic SFT for Cold Start.** We generate SFT demonstrations by rolling out a strong teacher policy on $(Q, A, \mathcal{T})$ tuples and apply rejection sampling: a trajectory is accepted if $\hat{A}$ matches $A$; otherwise, the task is discarded. The resulting dataset $\mathcal{D}_{\text{sft}} = \{(Q^{(i)}, A^{(i)}, \mathcal{T}^{(i)}, \tau^{(i)})\}$ consists of tasks with verified trajectories.

**Agentic RL.** Starting from the SFT checkpoint, we further optimize the agent's policy in the same toolset $\mathcal{T}$ to enhance robustness. Before RL updates, we estimate task learnability via self-sampling: for each task we generate $k_{\text{rl}}$ rollouts under $\mathcal{T}$ and score their final answers against $A$. We then filter

for *frontier tasks* where the success rate falls within a learnable range, yielding a dataset $\mathcal{D}_{\text{rl}} = \{(Q^{(i)}, A^{(i)}, \mathcal{T}^{(i)})\}_{i=1}^{N_{\text{rl}}}$. During RL, the policy interacts with tools to produce a final answer $\hat{A}$ and receives a composite reward

$$R = \alpha R_{\text{format}} + R_{\text{correct}}, \qquad (4)$$

where $R_{\text{correct}}$ reflects the correctness of $\hat{A}$ w.r.t. $A$, and $R_{\text{format}}$ penalizes invalid tool calls.

## 4. Experiments

To investigate whether diversity scaling during synthesis translates to broad generalization, we evaluate DIVE across 9 benchmarks spanning diverse tasks and toolsets. We detail our experimental setup and multi-level benchmark taxonomy (§4.1) and present main results in §4.2.

### 4.1. Experimental Setup

**DIVE Synthesis Details.** We instantiate both the *evidence collector* and *task generator* with Claude-4-Sonnet (Anthropic, 2025). Each synthesis cycle samples a configuration: a seed concept, a 15–50 tool subset (randomly shuffled), and 3–5 query exemplars. The evidence collector performs up to 6 tool-calling steps per iteration; the task generator derives a grounded QA pair in a single reasoning pass. We run $K=3$ collection–derivation iterations per cycle. All tool executions are performed against live tools.

**Training Details.** We use **Qwen3-8B** (Yang et al., 2025) as our backbone. **(a) SFT:** From a pool of 114k tasks, we use GPT-OSS-120B (Agarwal et al., 2025) as teacher to collect 48k trajectories (with rejection sampling) for fine-tuning (300 steps, batch size 64, learning rate 1e-5, max context 65,536 tokens, up to 50 tool-call turns), producing **DIVE-8B (SFT)**. **(b) RL:** From a separate pool of 38k tasks, we select 3.2k frontier tasks (1–5 successes in pass@8 self-sampling) and train with GRPO (Shao et al., 2024) (100 steps, batch size 512, learning rate 5e-6, max context 131,072 tokens, up to 100 tool-call turns), producing **DIVE-8B (RL)**.

**Benchmark suites.** We evaluate DIVE across three tiers: in-domain (L1) and two OOD settings distinguished by tool pool: general-purpose tools (L2) vs. specialized toolsets (L3). See Table 1 for details.

- **L1 (In-distribution Tasks):** 800 tasks with disjoint seed concepts but the same 373-tool pool. Each task samples a novel tool subset (15–50 tools).
- **L2 (OOD Tasks w/ General Tools):** Benchmarks using Search/Browse/Code Execution. This includes *General DeepResearch* tasks (GAIA (Mialon et al., 2023), HLE (Phan et al., 2025), BROWSECOMP (Wei et al., 2025), XBENCH (Chen et al., 2025)) and *Domain Deep-Research* tasks (FINSEARCHCOMP (Hu et al., 2025)). We use the 103-sample text-only validation subset for GAIA

*Table 1.* **Benchmark taxonomy and OOD factors w.r.t. DIVE training data.** L2 benchmarks use general-purpose tools; L3 benchmarks require specialized toolsets. **OOD Factors**: *Task*=shifted task distribution, *Pool*=unseen tool pool, *Set*=unseen toolset, *Proto*=non-OpenAI protocol, *Env*=stateful environment.

| Tier | Task Family | Benchmark | OOD Factors | Tool Pool | Toolset | Protocol | Env |
|------|-------------|-----------|-------------|-----------|---------|----------|-----|
| L1 | In-Distribution | DIVE-EVAL | – | 384 Tools (General + Expert) | Per-task | OpenAI | Stateless |
| L2 | General DeepResearch | GAIA, HLE, BROWSECOMP, XBENCH-DS | *Task, Set* | Search / Browse | Uniform | OpenAI | Stateless |
|    | Domain DeepResearch | FINSEARCHCOMP (Global) | *Task, Set* | Search / Browse / Code Execution | Uniform | OpenAI | Stateless |
| L3 | Financial Specialist | FINANCE AGENT BENCHMARK | *Task, Pool, Set* | EDGAR / Web / Parse / Retrieve | Uniform | OpenAI | Stateless |
|    | Medical Specialist | MEDAGENTBENCH | *Task, Pool, Set, Proto, Env* | FHIR GET / POST / Finish | Uniform | HTTP | Stateful |
|    | Software Engineering | SWE-BENCH VERIFIED | *Task, Pool, Set, Env* | Bash / Search / Editor / Finish | Uniform | OpenAI | Stateful |
|    | Zero-Shot Generalist | TOOLATHLON | *Task, Pool, Set, Env* | 604 Tools (32 MCP Apps) | Per-task | OpenAI | Stateful |

*Table 2.* **Overall comparison across L1–L3 benchmarks. L1**: in-distribution; **L2**: OOD w/ general tools; **L3**: OOD w/ specialized tools. BC=BrowseComp; XB-DS=Xbench-DeepSearch; FSC$_2$/FSC$_3$=FinSearchComp Global-T2/T3; FAB=Finance Agent Benchmark; MAB=MedAgentBench; SWE=SWE-bench Verified. 8B Baselines include specialized agentic models (WebExplorer-8B; our SWE-Dev-8B trained on SWE-Dev (Wang et al., 2025a)) and generalizable agentic models (EnvScaler-8B). Scores are success rates (%). Toolathlon is averaged over 3 runs; all other benchmarks are averaged over 4 runs. Underline: best overall; **Bold**: best among 8B backbone.

| Category | Model | L1 In-distribution | L2 OOD w/ General Tools | | | | | | L3 OOD w/ Specialized Tools | | | |
|----------|-------|--------------------|-------------------------|---|---|---|---|---|------------------------------|---|---|---|
|          |       | DIVE-Eval | GAIA | HLE | BC | XB-DS | FSC$_2$ | FSC$_3$ | FAB | MAB | SWE | Toolathlon |
| **Frontier** (≫**8B**) | Gemini-3-Pro | 45.3 | 80.3 | 42.9 | 49.0 | 76.0 | 70.6 | 52.4 | 39.0 | 74.8 | 76.2 | 36.4 |
|          | Claude-4-Sonnet | 44.8 | 63.7 | 20.8 | 12.8 | 62.2 | 60.2 | 33.3 | 39.0 | 79.3 | 72.7 | 29.9 |
|          | Gemini-2.5-Pro | 29.1 | 60.2 | 28.4 | 9.9 | 56.0 | 44.5 | 27.4 | 24.0 | 65.1 | 59.6 | 10.5 |
|          | DeepSeek-V3.2-Exp | 40.4 | 61.0 | 17.9 | 40.1 | 67.2 | 61.3 | 27.4 | 26.0 | 67.3 | 67.8 | 20.1 |
|          | Kimi-K2-0905 | 32.9 | 60.0 | 26.9 | 14.1 | 61.0 | 47.1 | 10.7 | 28.0 | 61.2 | 69.2 | 13.0 |
|          | GPT-OSS-120B | 40.5 | 66.0 | 19.0 | 27.0 | 69.5 | 61.0 | 22.0 | 34.0 | 64.3 | 62.0 | 9.8 |
| **8B Baselines** | WebExplorer-8B | 19.1 | 50.0 | 17.3 | 15.7 | 53.7 | 35.9 | 18.1 | 4.0 | 17.8 | 7.0 | 0.3 |
|          | SWE-Dev-8B | 13.8 | 23.2 | 6.9 | 1.6 | 31.6 | 30.5 | 3.1 | 3.0 | 14.2 | **19.5** | 0.0 |
|          | EnvScaler-8B | 15.4 | 25.8 | 2.8 | 1.7 | 45.7 | 40.7 | 10.8 | 14.0 | 56.6 | 11.5 | 2.2 |
| **Ours** | Qwen3-8B (base) | 13.0 | 22.4 | 6.4 | 1.3 | 24.0 | 28.6 | 7.1 | 2.0 | 38.4 | 10.8 | 0.9 |
|          | DIVE-8B (SFT) | 35.4 | 49.3 | 13.8 | 12.9 | 50.2 | 62.1 | 33.0 | 28.0 | 50.2 | 13.2 | 4.7 |
|          | DIVE-8B (RL) | **42.5** | **61.2** | **17.8** | **16.4** | **58.1** | **67.3** | **37.3** | **34.0** | **57.3** | 18.3 | **8.3** |

and the DeepSearch subset for XBENCH.

- **L3 (OOD Tasks w/ Specialized Tools):** Benchmarks requiring specialized toolsets, including FINANCE AGENT BENCHMARK (public validation set) (Choi et al., 2025) (financial APIs), MEDAGENTBENCH (Jiang et al., 2025) (EHR system), SWE-BENCH VERIFIED (Jimenez et al., 2023) (containerized codebase interaction), and TOOLATHLON (Li et al., 2025b) (diverse MCP toolsets).

**OOD Factors.** We categorize distribution shifts into five dimensions relative to DIVE's training data (Table 1). **Task distribution**: the variety of user instructions; a shift means evaluation on tasks not synthesized by DIVE (e.g., human-curated benchmarks). **Tool pool**: the benchmark-level universe of tool types; a shift involves unseen tools. **Toolset**: the task-level subset of tools; a shift tests unseen tool combinations. **Protocol**: the invocation interface (e.g., function-calling vs. raw HTTP); a shift requires adapting to different schemas. **Environment**: the execution substrate; a shift introduces stateful dynamics (e.g., Docker containers).

**Baselines.** We compare DIVE-8B against two categories of models: (i) **8B baselines** (same backbone trained on other synthesized data), including specialized models for *specific tasks/tools* (WebExplorer-8B (for general DeepResearch) (Liu et al., 2025b), SWE-Dev-8B trained on

the open-source SWE-Dev dataset (Wang et al., 2025a)) and generalizable models via *query-first synthesis* in simulated environments (EnvScaler-8B (Song et al., 2026)); (ii) **Frontier models** (≫8B), including Gemini-3-Pro (Google DeepMind, 2025), Claude-4-Sonnet (Anthropic, 2025), Gemini-2.5-Pro (Comanici et al., 2025), DeepSeek-V3.2-Exp (DeepSeek-AI, 2025), Kimi-K2-0905 (Team et al., 2025), and GPT-OSS-120B (Agarwal et al., 2025). For all models, we evaluate with temperature $= 1$ and top-$p = 1$.

### 4.2. Main Results

Table 2 presents the main evaluation results. In this section, we discuss the effectiveness of DIVE across diverse benchmarks and highlight the following observations:

**DIVE Delivers Robust and Substantial Generalization.** It improves on in-distribution tasks (L1) and transfers consistently to all OOD benchmarks spanning both general and specialized toolsets (Table 1). Across the 9 OOD benchmarks, DIVE improves by +16.2 (SFT) and +22.2 (RL) points per benchmark on average, outperforming the strongest 8B baseline by +68% (Table 2). Despite its small backbone, DIVE is competitive with much larger models on deep-research and on challenging specialized benchmarks (e.g., FAB and MAB; Table 2). Notably, TOOLATHLON

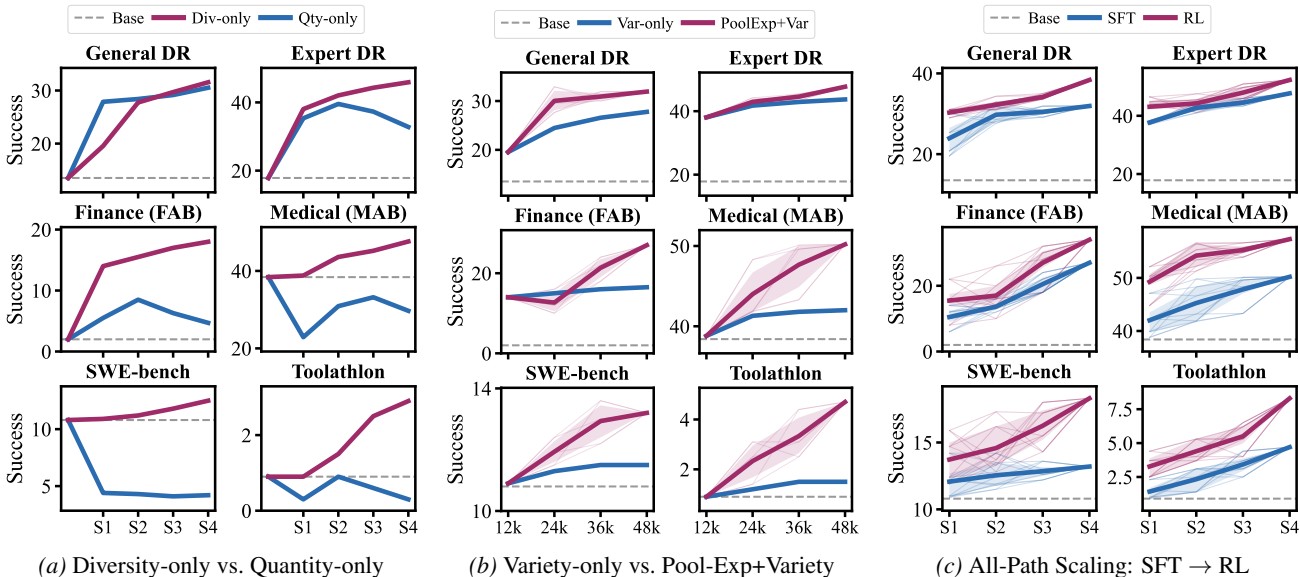

*Figure 3.* **Scaling analysis.** Gray dashed line: Qwen3-8B base. **Left: Diversity-only vs. Quantity-only.** Diversity-only expands the tool pool from 1→4 domains (12k fixed; representative path fin→fin+med→fin+med+bio→all). Quantity-only scales data 12k→48k with tasks/tools fixed (Gen-DR; Search/Browse-only); diversity yields stronger OOD gains. **Middle: Toolset-variety-only vs. Pool-Expansion+Variety.** Both scale SFT data 12k→48k from Finance. Toolset-variety-only: pool fixed. Pool-Expansion+Variety: pool expands across domains (multiple paths); pool expansion sustains gains. **Right: All-path scaling (SFT → RL).** 24 domain-expansion permutations; SFT 12k→48k and RL 0.8k→3.2k. Thin: paths; thick: mean; shaded: interquartile range; RL amplifies scaling.

is a stringent zero-shot benchmark with per-task MCP app toolsets and stateful container environments, where DIVE improves from near-zero to 8.3 points, approaching GPT-OSS-120B and Gemini-2.5-Pro.

**DIVE Wins by Generalization, Even Against Specialists.** Without task-specific training, DIVE matches or surpasses specialist agents on their home benchmarks (e.g., GAIA: 61.2 vs. 50.0 for WebExplorer-8B). In contrast, these specialists transfer poorly under unseen shifts, often exhibiting negative transfer (e.g., WebExplorer-8B drops 5.8 points below the base model on L3 benchmarks). Compared to other generalization-oriented synthesis baselines (e.g., EnvScaler-8B), DIVE achieves a 3.2× larger OOD lift, validating evidence-first synthesis on diverse, real tools.

## 5. Analysis

Our main results show that DIVE generalizes broadly across shifts in tasks and toolsets. We now ask what drives these gains and how they scale. In our method, we controllably scale *tool-pool coverage* and *toolset variety*, and further induce richer tool-use patterns through our synthesis loop (cf. §3.3). In §5.1, we systematically study how these controllable diversity axes shape the scaling trend of OOD generalization; in §5.2, we further analyze the structural diversity induced by DIVE to explain this trend.

### 5.1. Scaling Analysis

In this section, we systematically study the scaling trend of OOD generalization under tool shifts by controllably varying *tool-pool coverage* and *toolset variety* (Fig. 3). Specifically, we study the following three questions:

**How Necessary and Efficient Is Diversity Scaling for Generalizable Tool Use?** We run an *extreme* scaling comparison under matched synthesis and SFT settings (Fig. 3a). Diversity-only holds the total data budget fixed (12k) and scales tool-pool coverage from 1→4 domains, inducing richer tool-use patterns. Quantity-only holds the task and tool distribution fixed (Gen-DR: deep-research tasks synthesized with fixed Search/Browse tools) and scales data quantity from 12k→48k. The contrast is stark: diversity scaling yields consistent and stronger OOD gains, whereas Quantity-only scaling mainly improves in-distribution routines. Even with more data, quantity-only cannot close the generalization gap from missing diversity, and at larger scale can further widen it on most OOD benchmarks. Notably, with **4× less data** (12k vs. 48k), Diversity-only still consistently outperforms Quantity-only across benchmarks, helping explain the gap to narrow specialist baselines.

**How Should We Scale Diversity for Faster Gains and a Higher Generalization Ceiling?** We next ablate *how* to scale diversity in synthesis (Fig. 3b) with matched SFT scaling (12k→48k) starting from Finance. *Toolset-variety-only* keeps the Finance tool pool fixed; scaling data increases

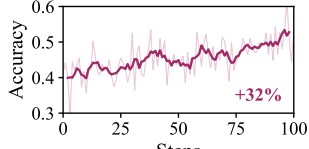 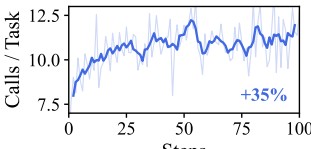 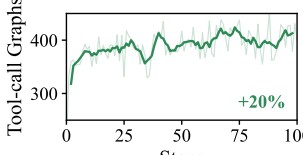 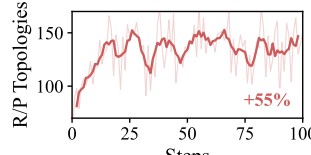

*Figure 4.* **RL training dynamics over 100 steps.** Accuracy reward, tool calls/task, unique tool-call graphs, and unique R/P topologies. Light: per-step; dark: smoothed. Percent changes use smoothed start/end values. Each step uses a 512-task RL batch, so per-step unique counts are upper-bounded by 512. Full 16-panel version in Appendix Figure 6.

distinct toolset variants, but the pool's tool *types* stay constant. *Pool-Expansion+Variety* similarly increases toolset variants, while also expanding the pool across domains to introduce new tool types. Our statistics show toolset-variant growth is similar under both routes; thus the key difference is whether new tool types/capabilities enter the pool. Empirically (Fig. 3b), both improve generalization, but toolset-variety-only yields **limited gains** and exhibits **faster saturation**, while Pool-Expansion+Variety yields **faster gains** and a **higher, slower-saturating ceiling** as the pool grows—making it a stronger scaling strategy.

**Does Exploration (RL) Further Amplify the Diversity-Scaling Trend Beyond Imitation (SFT)?** To robustly evaluate this trend, we evaluate SFT and RL across all 24 scaling paths (Fig. 3c; Table 13). The trend is already visible under SFT, indicating the model can *imitate* diverse tool-use patterns in expert trajectories. RL *amplifies* this diversity-scaling trend, suggesting exploration beyond imitation: the RL–SFT gap grows with diversity (Avg$_{RL}$−Avg$_{SFT}$: +4.6 at 1 domain vs. +5.6 at 4 domains), and at 4 domains the mean rises from 29.1→34.8 with narrow interquartile bands. In §5.2, we quantify structural diversity induced by DIVE to validate and help explain this mechanism.

### 5.2. Structural Diversity Analysis

At this stage, we analyze the **structural diversity** of tool use in DIVE's SFT trajectories and RL rollouts to explain the diversity-scaling trend in OOD generalization. We measure diversity at three levels: (i) **tool-pool coverage** (tool types exercised, e.g., *tools covered* in the dataset and *distinct tool types per task*); (ii) **toolset variety** (*unique toolsets* across tasks); and (iii) **tool-use patterns**, including *unique tool-call sequences*, *unique tool-call graphs* capturing tool–reasoning dependencies (inferred by Claude-4-Sonnet), and abstract *Retrieval/Processing (R/P) topologies* (222-class taxonomy; Appendix D). We compare DIVE's 48k SFT trajectories against Gen-DR (Table 3, Figure 5), then track how these patterns evolve during RL (Figure 4).

**SFT Data: Structural Diversity.** To interpret §5.1, we inspect SFT trajectories: DIVE exhibits higher diversity than Gen-DR in tool coverage, toolset variety, and tool-use

*Table 3.* **Diversity comparison: Gen-DR vs. DIVE** (48k each; same teacher model, rejection sampling).

| Diversity Metric | Gen-DR | DIVE | Δ |
|---|---|---|---|
| Tools covered | 2 | 373 | +186× |
| Unique toolsets | 1 | 46,398 | +46k× |
| Unique tool-call sequences | 1,231 | 25,084 | +20× |
| Unique tool-call graphs | 19,442 | 39,810 | +105% |
| Unique R/P topologies | 12,315 | 23,450 | +90% |
| R/P topology classes covered | 65 | 153 | +135% |
| Avg. tool calls per task | 15.21 | 11.11 | -27% |
| Distinct tool types per task | 1.71 | 3.26 | +91% |
| Avg. score after SFT | 22.51 | 32.15 | +43% |

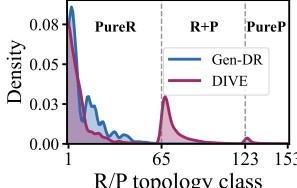 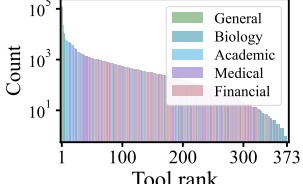

*Figure 5.* **R/P topology density and tool-frequency distributions (48k SFT). Left:** Density over R/P topology classes (153 observed; retrieval-only→mixed→processing-only, i.e., PureR→R+P→PureP; taxonomy in Appendix D). **Right:** Tool-call frequency over 373 tools (5 domains).

patterns (Table 3). Figure 5 highlights two key differences: in topology space (left), Gen-DR concentrates in retrieval-only (PureR), while DIVE shifts mass to mixed (R+P) and processing-only (PureP) and covers more topology classes; in tool usage (right), DIVE exhibits a long-tail frequency over a broad tool pool. These shifts match the OOD setting, where tasks often require retrieval–processing composition and specialized tools. Further, we analyze how synthesis-loop iterations increase diversity in Appendix C.

**RL Amplifies Diversity.** During RL, accuracy improves while structural diversity continues to increase (Figure 4). Over 100 RL iterations, reward improves while diverse tool-use patterns persist and expand (e.g., unique tool-call graphs and R/P topologies). See Appendix Figure 6 for a per-domain breakdown of these dynamics. Together with §5.1, this supports our hypothesis that RL amplifies generalization by exploring and reinforcing a broader set of effective tool-use structures rather than collapsing to a single routine.

## Conclusion

We presented DIVE, an execution-first framework for training tool-using agents on *real*, *diverse* toolsets with built-in executability and verifiability. By inverting synthesis (evidence first, tasks derived from traces), DIVE scales diversity while keeping supervision grounded. Across three benchmark tiers, DIVE improves OOD generalization, and scaling studies show *tool-pool diversity* matters more than data quantity. Structural analyses reveal richer tool-use patterns (sequences, graphs, R/P topologies), a trend visible under SFT and amplified by RL.

## Acknowledgements

We thank MiniMax for providing computational resources and infrastructure support, and the anonymous reviewers for their constructive feedback.

## Impact Statement

This work studies data synthesis and post-training for tool-using language agents. By generating grounded tasks and training on diverse, real-world tools, we aim to expand the supply of diverse agentic training data, improve generalization across tasks and toolsets, and enable more reliable agent evaluation.

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

# A. Synthesized Task Examples

We present representative examples of synthesized tasks from each domain to illustrate the complexity and diversity achieved by DIVE. Each task requires multi-step reasoning across multiple tools selected from a domain-specific candidate set.

In the examples below, the **Tools** field lists the full candidate toolset available for the task, where `green tags` indicate tools effectively used by the agent and `gray tags` indicate available but unused tools. This visualization highlights the agent's ability to precisely select relevant tools from a noisy candidate set to solve complex queries. The **Stats** line provides quantitative metrics: **Calls** (total tool calls), **Available** (size of toolset), and **Unique** (count of distinct tools used).

## A.1. Academic Domain

> **Academic Case: Cross-Journal Researcher Tracking**
>
> **Tools:** `arxiv_advanced_search` `arxiv_get_papers_by_ids` `arxiv_search_by_author` `arxiv_search_by_category` `arxiv_search_by_date_range` `arxiv_search_papers` `crossref_funder_works` `crossref_funders` `crossref_get_journal` `crossref_get_work` `crossref_journal_works` `crossref_journals` `crossref_licenses` `crossref_member_works` `crossref_work_agency` `openalex_get_author` `openalex_get_institution` `openalex_get_source` `openalex_get_work` `openalex_search_authors` `openalex_search_funders` `openalex_search_institutions` `openalex_search_publishers` `openalex_search_sources` `openalex_search_topics` `openalex_search_works`
>
> **Stats: Calls:** 50   **Available:** 26   **Unique:** 9
>
> **Query:** A Stanford University computer science researcher with an ORCID identifier ending in 3426 has exactly 260 publications cited at least 10 times each, and their most highly cited work was published in 1981. This researcher's total citation count exceeds 65,000 but is less than 66,000, and they have been affiliated with Stanford University continuously from 2014 through 2023. Among their publications from 2020-2023, what is the title of their work that appears in "ACM Transactions on Management Information Systems" and has been cited more than 40 times?
>
> **Answer:** PANDA: Partitioned Data Security on Outsourced Sensitive and Non-sensitive Data.

## A.2. Biological Domain

> **Biological Case: Enzyme Characterization with Taxonomic Constraints**
>
> **Tools:** `bio_seq_transcribe` `bio_sequtils_gc_skew` `bio_sequtils_nt_search` `bio_sequtils_seq1` `cluster_treecluster` `codon_table_standard` `expasy_prodoc` `expasy_prosite_raw` `ncbi_entrez_fetch` `ncbi_entrez_search` `ncbi_entrez_summary` `pairwise2_globalxx` `pairwise2_localxx` `protparam_aromaticity` `restriction_all_enzymes` `seqfeature_location` `svd_superimpose` `togows_convert` `togows_entry` `togows_search`
>
> **Stats: Calls:** 47   **Available:** 20   **Unique:** 6
>
> **Query:** I need the UniProt accession number for a thermostable cutinase that meets these interconnected criteria: (1) It originates from a bacterial species containing "fusca" in its scientific name and belongs to the same taxonomic family as the organism that produces BTA-hydrolase 2, (2) It has an alternative name containing "BTA-hydrolase" with the same numerical designation as the number of distinct BTA-hydrolase variants found in the NCBI protein database for this species, (3) It can degrade PET with EC number 3.1.1.101 and shares >95% sequence identity with CAH17553, (4) Its UniProt entry was most recently modified in the same month as the Zeikus thermozyme review publication but in 2025, (5) The protein must have the exact same number of amino acid residues as BTA-hydrolase 2 from the same organism, and (6) Its gene has a synonym that matches the primary gene name of its homolog minus one character.
>
> **Answer:** Q6A0I4

## A.3. Financial Domain

---

**Financial Case: Cross-Market Portfolio Analysis**

**Tools:** `browse`  `cb_call`  `cb_share`  `etf_daily`  `etf_index`  `film_record`  `fund_nav`  `fund_portfolio`  `fut_settle`  `fut_weekly_detail`  `index_sw_daily`  `jupyter_execute_code_cell`  `parallel_search`  `search`  `stock_balancesheet`  `stock_ccas_hold`  `stock_ccas_hold_detail`  `stock_disclosure_date`  `stock_dividend`  `stock_em_hot`  `stock_hsgt_top10`  `stock_kpl_topic`  `stock_limit_board`  `stock_managers`  `stock_stk_auction`  `stock_suspend`  `stock_tdx_index`  `tv_record`

**Stats: Calls:** 47    **Available:** 28    **Unique:** 8

**Query:** Among the fund holdings in portfolio 001753.OF as of December 31, 2023, identify the healthcare/medical diagnostics sector stock with the highest market value that also has a stock float ratio below 0.03 and was established before 2000. Then, cross-reference this with the Shanghai-Shenzhen-Hong Kong Stock Connect top 10 trading data for December 20, 2024, to find if any semiconductor industry stock with a trading amount exceeding 1.1 billion yuan shares the same first three digits in its stock code and has "innovation" in its Chinese company name. Additionally, verify that both companies are listed on the same exchange (Shanghai vs Shenzhen) and that the semiconductor company's English name contains "Semiconductor". What is the stock code of the fund holding that meets these industry-specific criteria, what is the English company name of its matching trading counterpart, and what year was the healthcare company established?

**Answer:** Stock code: 603658.SH; Chinese company name: GigaDevice Semiconductor (Beijing) Inc.; Healthcare company established: 1998

---

## A.4. Medical Domain

> ### Medical Case: Contraindication and Drug Classification Filtering
>
> **Tools:** `dbvar_germline_data` `genetic_diseases` `hcpcs_procedure_codes` `icd10cm_diagnosis_codes` `icd9cm_diagnosis_codes` `pharmvar_star_alleles` `prescribable_find_rxcui_by_id` `prescribable_find_rxcui_by_string` `prescribable_get_all_concepts_by_tty` `prescribable_get_drugs` `prescribable_get_prop_categories` `prescribable_get_related_by_type` `prescribable_get_rxnorm_name` `prescribable_get_source_types` `prescribable_get_spelling_suggestions` `prescribable_get_term_types` `rxclass_find_similar_classes_by_drug_list` `rxclass_get_all_classes` `rxclass_get_class_by_rxnorm_drug_name` `rxclass_get_class_contexts` `rxclass_get_class_members` `rxclass_get_class_tree` `rxclass_get_rela_source_version` `rxclass_get_relas` `rxclass_get_sources_of_drug_class_relations` `rxclass_get_spelling_suggestions` `rxnorm_find_related_ndcs` `rxnorm_find_rxcui_by_id` `rxnorm_find_rxcui_by_string` `rxnorm_get_all_properties` `rxnorm_get_all_related_info` `rxnorm_get_id_types` `rxnorm_get_ndc_status` `rxnorm_get_proprietary_information` `rxnorm_get_related_by_relationship` `rxnorm_get_rx_concept_properties` `rxnorm_get_rxcui_history_status` `rxnorm_get_source_types` `rxnorm_get_term_types` `rxterms_prescription_drugs`
>
> **Stats: Calls:** 44    **Available:** 40    **Unique:** 14
>
> **Query:** Among all drugs in the ATC therapeutic subgroup A10BK that share the same specific mechanism of action classification (MOA class N0000187058) and have contraindications with chronic kidney failure (MEDRT disease class D007676), identify which drug has UNII code 6C282481IP. For this drug: (1) calculate the numerical difference between its two available tablet strengths, (2) determine how many other active ingredients in the same MOA class have anhydrous or propanediol salt forms available in RxNorm, and (3) among the drugs in this MOA class that also have contraindications with chronic kidney failure, identify which one has the most brand name variations and state the total count of those brand names.
>
> **Answer:** Ertugliflozin (UNII 6C282481IP). (1) 10 MG (15 MG − 5 MG); (2) 2 other ingredients: canagliflozin anhydrous, dapagliflozin propanediol; (3) Empagliflozin with 4 brand names: Jardiance, Glyxambi, Synjardy, Trijardy.

# B. Data Synthesis Details

## B.1. Tool Pool Details

We curate a diverse tool pool spanning five domains with both **Retrieval (R)** and **Processing (P)** primitives. **Retrieval** tools fetch data from external sources (APIs, databases) without significant transformation. **Processing** tools perform calculations, analysis, or data transformations (e.g., technical indicators, sequence alignment, similarity matching).

*Table 4.* **Tool pool summary.** Distribution of Retrieval (R: information fetching) and Processing (P: computation/transformation) tools across domains.

| Domain | Retrieval | Processing | Total | P% |
|---|---|---|---|---|
| Financial | 155 | 14 | 169 | 8.3% |
| Medical | 80 | 9 | 89 | 10.1% |
| Academic | 43 | 7 | 50 | 14.0% |
| Biological | 18 | 43 | 61 | 70.5% |
| General | 3 | 1 | 4 | 25.0% |
| **Total** | **299** | **74** | **373** | **19.8%** |

*Table 5.* **Tool sources and documentation.** Official documentation or API endpoints for the tools used in DIVE.

| Domain | Source Name | URL / Documentation |
|---|---|---|
| Financial | Tushare | https://tushare.pro/ |
| Medical | RxNorm API | https://lhncbc.nlm.nih.gov/RxNav/APIs/RxNormAPIs.html |
| | RxClass API | https://lhncbc.nlm.nih.gov/RxNav/APIs/RxClassAPIs.html |
| | Clinical Tables | https://clinicaltables.nlm.nih.gov/ |
| Academic | Semantic Scholar | https://www.semanticscholar.org/product/api |
| | OpenAlex | https://docs.openalex.org/ |
| | Crossref | https://www.crossref.org/documentation/retrieve-metadata/rest-api/ |
| | arXiv | https://arxiv.org/help/api |
| Biological | NCBI Entrez | https://www.ncbi.nlm.nih.gov/books/NBK25501/ |
| | BioPython | https://biopython.org/ |
| | TogoWS | http://togows.dbcls.jp/ |
| General | Google Search | https://www.google.com/ |
| | Python Sandbox | Custom Implementation (based on standard library) |

The complete list of all 373 tools is provided in Table 6.

*Table 6.* **Complete tool pool (373 tools).** Aggregated by category. R/P denotes Retrieval (R) or Processing (P) tool type.

| Category | Domain | R/P | Count | Tools |
|---|---|---|---|---|
| **Conv. Bonds** | Financial | P | 1 | cb_factor_pro |
| **ETF Data** | Financial | P | 1 | etf_adj |
| **Mutual Funds** | Financial | P | 1 | fund_factor |
| **Indices** | Financial | P | 1 | index_factor |
| **Stock Metrics** | Financial | P | 10 | stock_adj_factor, stock_ah_ratio, stock_cyq_perf, stock_daily_basic, stock_fina_indicator, stock_fina_indicator_vip, stock_stk_ah_comparison, stock_stk_factor, stock_stk_factor_pro, stock_stk_nine_turn |
| **Bonds/Rates** | Financial | R | 7 | bc_bestotcqt, bc_otcqt, libor, repo_daily, us_tbr, us_trycr, us_tycr |
| **Conv. Bonds** | Financial | R | 6 | cb_basic, cb_call, cb_daily, cb_issue, cb_rate, cb_share |
| **ETF Data** | Financial | R | 3 | etf_basic, etf_daily, etf_index |
| **FX & HK** | Financial | R | 4 | fx_daily, hibor, hk_basic, hk_tradecal |
| **Mutual Funds** | Financial | R | 6 | fund_basic, fund_div, fund_manager, fund_nav, fund_portfolio, fund_share |
| **Futures** | Financial | R | 9 | fut_basic, fut_daily, fut_holding, fut_limit, fut_mapping, fut_settle, fut_weekly_detail, fut_weekly_monthly, fut_wsr |
| **Indices** | Financial | R | 12 | index_basic, index_ci_daily, index_ci_member, index_daily, index_daily_info, index_global, index_monthly, index_sw_daily, index_sw_member, index_sz_market, index_weekly, index_weight |
| **Macro/Other** | Financial | R | 13 | anns_d, cn_gdp, cn_m, cn_pmi, cn_ppi, film_record, news_cctv, sse_qa, szse_qa, tmt_twincome, tmt_twincome_detail, trade_cal, tv_record |
| **Options** | Financial | R | 2 | opt_basic, opt_daily |

*(continued from previous page)*

| Category | Domain | R/P | Count | Tools |
|---|---|---|---|---|
| **Stock Data** | Financial | R | 90 | stock_auction_close, stock_auction_open, stock_bak_daily, stock_balancesheet, stock_balancesheet_vip, stock_block_trade, stock_broker_forecast, stock_broker_recommend, stock_broker_recommend, stock_cashflow, stock_cashflow_vip, stock_ccas_hold, stock_ccas_hold_detail, stock_ccass_hold, stock_ccass_hold_detail, stock_company, stock_concept_detail, stock_daily, stock_dc_daily, stock_dc_index, stock_dc_member, stock_disclosure_date, stock_dividend, stock_em_hot, stock_express, stock_fina_mainbz, stock_forecast, stock_ggt_daily, stock_ggt_monthly, stock_ggt_top10, stock_hk_hold, stock_hm_detail, stock_hm_list, stock_holder_number, stock_hs_const, stock_hsgt_top10, stock_income, stock_income_vip, stock_index_member, stock_kpl_list, stock_kpl_topic, stock_lhb_detail, stock_limit_board, stock_limit_list_d, stock_limit_list_ths, stock_limit_step, stock_managers, stock_margin, stock_margin_detail, stock_market_money_flow, stock_money_flow, stock_moneyflow, stock_moneyflow_hsgt, stock_moneyflow_ths, stock_monthly, stock_namechange, stock_pledge_detail, stock_pledge_stat, stock_pro_bar, stock_report_rc, stock_repurchase, stock_sector_money_flow, stock_share_float, stock_slb_len_mm, stock_slb_sec, stock_slb_sec_detail, stock_stk_auction, stock_stk_holdertrade, stock_stk_holds, stock_stk_limit, stock_stk_limit_pool, stock_stk_rewards, stock_stk_splits, stock_stk_surv, stock_stk_surv, stock_suspend, stock_suspend_d, stock_tdx_daily, stock_tdx_index, stock_tdx_member, stock_ths_daily, stock_ths_hot, stock_ths_index, stock_ths_member, stock_top10_floatholders, stock_top10_holders, stock_top_inst, stock_top_list, stock_trade_cal, stock_weekly |
| **Stock Metrics** | Financial | R | 5 | stock_bak_basic, stock_basic, stock_cyq_chips, stock_cyq_perc, stock_index_dailybasic |
| **Matching** | Medical | P | 8 | prescribable_get_approximate_match, prescribable_get_spelling_suggestions, rxclass_find_similar_classes_by_class, rxclass_find_similar_classes_by_drug_list, rxclass_get_similarity_information, rxclass_get_spelling_suggestions, rxnorm_get_approximate_match, rxnorm_get_spelling_suggestions |
| **Prescribable** | Medical | P | 1 | prescribable_find_rxcui_by_string |
| **Prescribable** | Medical | R | 21 | prescribable_filter_by_property, prescribable_find_rxcui_by_id, prescribable_get_all_concepts_by_tty, prescribable_get_all_properties, prescribable_get_all_related_info, prescribable_get_display_terms, prescribable_get_drugs, prescribable_get_id_types, prescribable_get_multi_ingred_brand, prescribable_get_ndcs, prescribable_get_prop_categories, prescribable_get_prop_names, prescribable_get_rela_paths, prescribable_get_rela_types, prescribable_get_related_by_relationship, prescribable_get_related_by_type, prescribable_get_rx_concept_properties, prescribable_get_rx_property, prescribable_get_rxnorm_name, prescribable_get_source_types, prescribable_get_term_types |
| **Reference** | Medical | R | 14 | cytogenetic_chromosome_locations, dbvar_germline_data, genetic_diseases, hcpcs_procedure_codes, hugo_genes, icd10cm_diagnosis_codes, icd9cm_diagnosis_codes, pharmvar_star_alleles, reference_sequences, rxterms_get_all_concepts, rxterms_get_all_rxterm_info, rxterms_get_rxterm_display_name, rxterms_get_rxterms_version, rxterms_prescription_drugs |
| **RxClass** | Medical | R | 13 | rxclass_find_class_by_name, rxclass_find_classes_by_id, rxclass_get_all_classes, rxclass_get_class_by_rxnorm_drug_id, rxclass_get_class_by_rxnorm_drug_name, rxclass_get_class_contexts, rxclass_get_class_graph_by_source, rxclass_get_class_members, rxclass_get_class_tree, rxclass_get_class_types, rxclass_get_rela_source_version, rxclass_get_relas, rxclass_get_sources_of_drug_class_relations |
| **RxNorm** | Medical | R | 32 | rxnorm_filter_by_property, rxnorm_find_related_ndcs, rxnorm_find_rxcui_by_id, rxnorm_find_rxcui_by_string, rxnorm_get_all_concepts_by_status, rxnorm_get_all_concepts_by_tty, rxnorm_get_all_historical_ndcs, rxnorm_get_all_ndcs_by_status, rxnorm_get_all_properties, rxnorm_get_all_related_info, rxnorm_get_display_terms, rxnorm_get_drugs, rxnorm_get_id_types, rxnorm_get_multi_ingred_brand, rxnorm_get_ndc_properties, rxnorm_get_ndc_status, rxnorm_get_ndcs, rxnorm_get_prop_categories, rxnorm_get_prop_names, rxnorm_get_proprietary_information, rxnorm_get_reformulation_concepts, rxnorm_get_rela_paths, rxnorm_get_rela_types, rxnorm_get_related_by_relationship, rxnorm_get_related_by_type, rxnorm_get_rx_concept_properties, rxnorm_get_rx_property, rxnorm_get_rxcui_history_status, rxnorm_get_rxnorm_name, rxnorm_get_rxnorm_version, rxnorm_get_source_types, rxnorm_get_term_types |

*(continued from previous page)*

| Category | Domain | R/P | Count | Tools |
|---|---|---|---|---|
| **Analysis** | Academic | P | 4 | openalex_analyze_text, semantic_scholar_paper_autocomplete, semantic_scholar_paper_recommendations, semantic_scholar_recommend_papers |
| **S. Scholar** | Academic | P | 3 | semantic_scholar_paper_search, semantic_scholar_paper_title_search, semantic_scholar_snippet_search |
| **Crossref** | Academic | R | 15 | crossref_funder_works, crossref_funders, crossref_get_funder, crossref_get_journal, crossref_get_member, crossref_get_prefix, crossref_get_type, crossref_get_work, crossref_journal_works, crossref_journals, crossref_licenses, crossref_member_works, crossref_members, crossref_types, crossref_work_agency |
| **OpenAlex** | Academic | R | 11 | openalex_get_author, openalex_get_institution, openalex_get_source, openalex_get_work, openalex_search_authors, openalex_search_funders, openalex_search_institutions, openalex_search_publishers, openalex_search_sources, openalex_search_topics, openalex_search_works |
| **S. Scholar** | Academic | R | 11 | semantic_scholar_author_batch, semantic_scholar_author_papers, semantic_scholar_author_search, semantic_scholar_get_author, semantic_scholar_get_paper, semantic_scholar_paper_authors, semantic_scholar_paper_batch, semantic_scholar_paper_bulk_search, semantic_scholar_paper_citations, semantic_scholar_paper_references, semantic_scholar_release_list |
| **arXiv** | Academic | R | 6 | arxiv_advanced_search, arxiv_get_papers_by_ids, arxiv_search_by_author, arxiv_search_by_category, arxiv_search_by_date_range, arxiv_search_papers |
| **Alignment** | Biological | P | 4 | pairwise2_global_align, pairwise2_globalxx, pairwise2_local_align, pairwise2_localxx |
| **Clustering** | Biological | P | 3 | cluster_distancematrix, cluster_pca, cluster_treecluster |
| **Motifs** | Biological | P | 3 | motifs_create, motifs_reverse_complement, motifs_reverse_complement_rna |
| **NCBI** | Biological | P | 2 | ncbi_entrez_ecitmatch, ncbi_entrez_espell |
| **Other** | Biological | P | 2 | svd_superimpose, togows_convert |
| **PDB** | Biological | P | 4 | pdb_calc_angle, pdb_calc_dihedral, pdb_is_aa, pdb_is_nucleic |
| **Protein** | Biological | P | 4 | protparam_analysis, protparam_aromaticity, protparam_isoelectric_point, protparam_molecular_weight |
| **Restriction** | Biological | P | 2 | restriction_catalyse, restriction_search |
| **Features** | Biological | P | 2 | seqfeature_compound, seqfeature_location |
| **Sequence** | Biological | P | 17 | bio_seq_back_transcribe, bio_seq_complement, bio_seq_complement_rna, bio_seq_count, bio_seq_find, bio_seq_pattern, bio_seq_reverse_complement, bio_seq_reverse_complement_rna, bio_seq_transcribe, bio_seq_translate, bio_sequtils_gc123, bio_sequtils_gc_content, bio_sequtils_gc_skew, bio_sequtils_nt_search, bio_sequtils_seq1, bio_sequtils_seq3, bio_sequtils_six_frame |
| **NCBI** | Biological | R | 5 | ncbi_entrez_einfo, ncbi_entrez_elink, ncbi_entrez_fetch, ncbi_entrez_search, ncbi_entrez_summary |
| **Other** | Biological | R | 11 | codon_table_by_id, codon_table_list, codon_table_standard, expasy_prodoc, expasy_prosite, expasy_prosite_raw, iupac_data_letters, iupac_data_weights, togows_entry, togows_search, togows_search_count |
| **Restriction** | Biological | R | 2 | restriction_all_enzymes, restriction_enzyme_info |
| **General** | General | P | 1 | code_execution |
| **General** | General | R | 3 | browse, parallel_search, search |

## B.2. Exemplar Sources

## B.3. Synthesis Prompts

We provide the core prompts used in the DIVE synthesis pipeline. Variable names in brackets (e.g., `{domain}`) are placeholders filled during runtime.

*Table 7.* **Exemplar sources.** We sample 3,000 tasks from the following agentic benchmarks to serve as structural priors for task synthesis. These benchmarks are selected for their alignment with DIVE's focus on domain-specific knowledge, multi-hop retrieval, and complex processing.

| Benchmark | Task Description |
| --- | --- |
| WebArena (Zhou et al., 2023) | Long-horizon realistic web information seeking |
| AgentBench (Liu et al., 2023) | Comprehensive evaluation across multiple environments |
| ToolBench (Qin et al., 2023) | Diverse instruction following with real-world APIs |
| DSBench (Jing et al., 2024) | Data analysis and SQL/Python code generation |
| BrowseComp (Wei et al., 2025) | Web navigation and information extraction |
| Mind2Web (Deng et al., 2023) | Generalizable web interactions across domains |
| HotpotQA (Yang et al., 2018) | Multi-hop information retrieval and reasoning |
| GAIA (Mialon et al., 2023) | Complex multi-step reasoning and tool planning |
| HLE (Phan et al., 2025) | Deep multidisciplinary reasoning and knowledge |
| FinQA (Chen et al., 2021) | Numerical reasoning over financial reports |
| TAT-QA (Zhu et al., 2021) | Hybrid reasoning over tabular and textual data |
| PubMedQA (Jin et al., 2019) | Biomedical research question answering |
| BioASQ (Tsatsaronis et al., 2015) | Biomedical semantic indexing and question answering |
| API-Bank (Li et al., 2023) | Tool usage and dialogue management |
| Frames (Krishna et al., 2025) | Unified evaluation of retrieval-augmented generation |
| SciCode (Tian et al., 2024) | Research-level scientific coding problems |
| $\tau$-bench (Yao et al., 2024) | Tool-agent-user interaction in real-world domains |
| TheAgentCompany (Xu et al., 2024) | Consequential real-world professional tasks |
| TravelPlanner (Xie et al., 2024) | Real-world travel planning with constraints |
| LegalAgentBench (Li et al., 2025a) | Legal reasoning and document analysis with tools |
| HealthBench (Arora et al., 2025) | Comprehensive health-related LLM evaluation |
| MCP-Bench (Wang et al., 2025b) | Complex real-world tasks via MCP servers |

**Evidence Collection.** The agent interacts with the sampled toolset to accumulate grounded evidence (tool execution traces with outputs).

---

**Evidence Collection (Round 1)**

```
Research "{seed_concept}" in {domain} domain.  Use multiple tools to retrieve and
process comprehensive and verifiable information from various sources.
Step budget:  {max_steps}

Note:  Investigate the topic from multiple angles and explore its connections to
related entities or concepts.

Strategy:  If direct search has limited results, try related concepts, broader
categories, or alternative terms.  Consider how the different aspects of the topic
relate to each other.
```

---

**Evidence Collection (Round $K > 1$)**

```
Continue research on "{current_query}" in {domain} domain.
Step budget:  {max_steps}

Previous findings:
{accumulated_evidence}

Based on previous findings, expand the research to broader or deeper aspects.  Use
diverse tools to retrieve and process new information.  Avoid repeating previous
findings.
```

---

**Task Derivation.** Based on the accumulated evidence, the model derives a query-answer pair strictly grounded in the execution traces.

---

**Task Derivation (Round 1)**

```
Exemplars:
{exemplars}

Seed:  {seed_concept}
Evidence collected:  {accumulated_evidence}

Derive a specific and realistic query using the collected data.  Base the answer on
actual tool results only.

QUERY: [specific query grounded in evidence]
ANSWER: [concise factual answer from tool results only - no explanations, no reasoning,
just the key values/facts]
REASONING: [how the evidence supports this query-answer pair]
```

---

**Task Derivation (Round $K > 1$)**

```
Exemplars:
{exemplars}

Current:  {current_query}
Evidence collected:  {accumulated_evidence}

Refine the question to be more challenging, specific and realistic using the diverse
collected data.  Base answer on actual tool results only.

EVOLVED_QUERY: [more complex question using collected data]
EVOLVED_ANSWER: [brief, factual answer from tool results - be concise, specific to the
question]
REASONING: [what complexity was added]
```

**Verification.** We use Claude-4-Sonnet and DeepSeek-V3.2 as cross-verifiers; an answer is marked correct only if both models agree. On a 200-sample human audit, verifiers achieved 100% agreement, owing to concise, unambiguous reference answers.

---

**Answer Verification**

```
Evaluate the correctness of the model's answer.

QUERY: {query}
REFERENCE ANSWER: {ground_truth}
MODEL ANSWER: {model_response}

Evaluation criteria:
- Compare factual content, not surface format
- Ignore differences in phrasing or presentation
- Focus on whether the core factual claims are correct

Output format:
JUDGEMENT: [correct/partial/incorrect]
EXPLANATION: [Brief justification]

Use "correct" if all key facts match, "partial" if the core answer is right but some
details are wrong or missing, "incorrect" if the main answer is wrong.
```

## C. Diversity Analysis

To analyze how iterative synthesis increases task diversity, we sample 4,000 tasks and evaluate structural diversity metrics across iteration rounds. For each iteration count $K \in \{1, 2, 3\}$, we use GPT-OSS-120B to solve the synthesized tasks and measure diversity in the resulting trajectories.

*Table 8.* **Structural diversity across iteration rounds.** We sample 4,000 tasks and measure diversity metrics in trajectories generated by GPT-OSS-120B. All diversity metrics increase substantially with more iterations, while pass rate decreases (indicating harder tasks).

| Metric | K=1 | K=2 | K=3 | $\Delta$ (1$\rightarrow$3) |
|---|---|---|---|---|
| GPT-OSS-120B Pass Rate (%) | 66.8 | 51.1 | 41.2 | −38% |
| Avg. Tool Calls per Task | 4.6 | 7.5 | 10.1 | +120% |
| Tool Coverage (%) | 58.2 | 74.3 | 92.2 | +58% |
| Unique Tool-call Sequences | 892 | 1,321 | 2,348 | +163% |
| Unique Tool-call Graphs | 1,362 | 2,467 | 3,551 | +161% |
| Unique R/P Topologies | 542 | 1,004 | 2,393 | +341% |
| R/P Topology Classes | 42 | 121 | 165 | +293% |
| Distinct Tools per Task | 1.89 | 2.24 | 3.15 | +67% |

**Key observations.** (1) **Diversity scales with iterations**: All structural diversity metrics increase substantially from $K=1$ to $K=3$. Tool coverage grows from 58.2% to 92.2%, unique R/P topologies increase by 341%, and topology class coverage expands from 42 to 165 classes. (2) **Task complexity increases**: The decreasing pass rate (66.8%$\rightarrow$41.2%) and increasing tool calls per task (4.6$\rightarrow$10.1) indicate that iterative synthesis produces more challenging tasks requiring longer solution trajectories. (3) **Richer tool compositions**: Distinct tools per task grows from 1.89 to 3.15, showing that later iterations induce tasks requiring more diverse tool combinations rather than repetitive single-tool patterns.

These results validate the iterative synthesis design: each additional iteration explores new regions of the tool space, producing tasks that cover more tools, exhibit more structural variety, and require more sophisticated reasoning.

## D. Topology Class Definition

We define topology classes using a 3-level hierarchy to systematically categorize tool-call graph patterns. Table 9 summarizes the classification scheme.

*Table 9.* Topology class dimensions (3-level hierarchy). Structure types are checked in priority order and are mutually exclusive.

| Level | Dimension | Values | Definition |
|---|---|---|---|
| 1. R/P Type | PureR | – | Only Retrieval tools |
| | R+P | – | Both Retrieval and Processing |
| | PureP | – | Only Processing tools |
| 2. Structure (priority order) | Single | $n = 1$ | Single tool call |
| | Indep | $n > 1 \wedge |E| = 0$ | Multiple independent calls |
| | Phain | $e = n-1 \wedge \max(\text{in/out}) \leq 1$ | Linear chain |
| | Fork | sources $= 1 \wedge$ sinks $> 1 \wedge \max(\text{in}) \leq 1$ | One-to-many |
| | Join | sinks $= 1 \wedge$ sources $> 1 \wedge \max(\text{out}) \leq 1$ | Many-to-one |
| | DAG | $\max(\text{in}) > 1 \wedge \max(\text{out}) > 1$ | Complex DAG |
| | Mix | Other | Mixed structure |
| 3. Scale | Depth (Phain/Fork/Join/DAG/Mix) | d1-2, d3-4, d5-7, d8+ | Longest path length |
| | Width (Fork/Join/DAG/Mix) | w1-2, w3-5, w6-10, w11+ | Max BFS layer width |
| | Node count (Indep only) | n2-3, n4-6, n7-10, n11-20, n21+ | Number of independent calls |

*Table 10.* Theoretical class count per structure type ($\times$ 3 R/P types).

| Structure | Scale Params | Bins | Classes |
|---|---|---|---|
| Single | none | 1 | $3 \times 1 = 3$ |
| Indep | node count | 5 | $3 \times 5 = 15$ |
| Phain | depth only | 4 | $3 \times 4 = 12$ |
| Fork | depth $\times$ width | $4 \times 4$ | $3 \times 16 = 48$ |
| Join | depth $\times$ width | $4 \times 4$ | $3 \times 16 = 48$ |
| DAG | depth $\times$ width | $4 \times 4$ | $3 \times 16 = 48$ |
| Mix | depth $\times$ width | $4 \times 4$ | $3 \times 16 = 48$ |
| **Total** | | | **222** |

**Naming convention:** Format varies by structure type:

- Single: `{R/P}/Single`, e.g., `PureR/Single`
- Indep: `{R/P}/Indep/{n}`, e.g., `R+P/Indep/n4-6`
- Phain: `{R/P}/Phain/{d}`, e.g., `PureP/Phain/d3-4`
- Fork/Join/DAG/Mix: `{R/P}/{Structure}/{d}/{w}`, e.g., `R+P/DAG/d5-7/w3-5`

DIVE covers 153 of 222 possible classes (69%); Gen-DR covers only 65 (all PureR, 29%).

# E. Additional Experimental Results

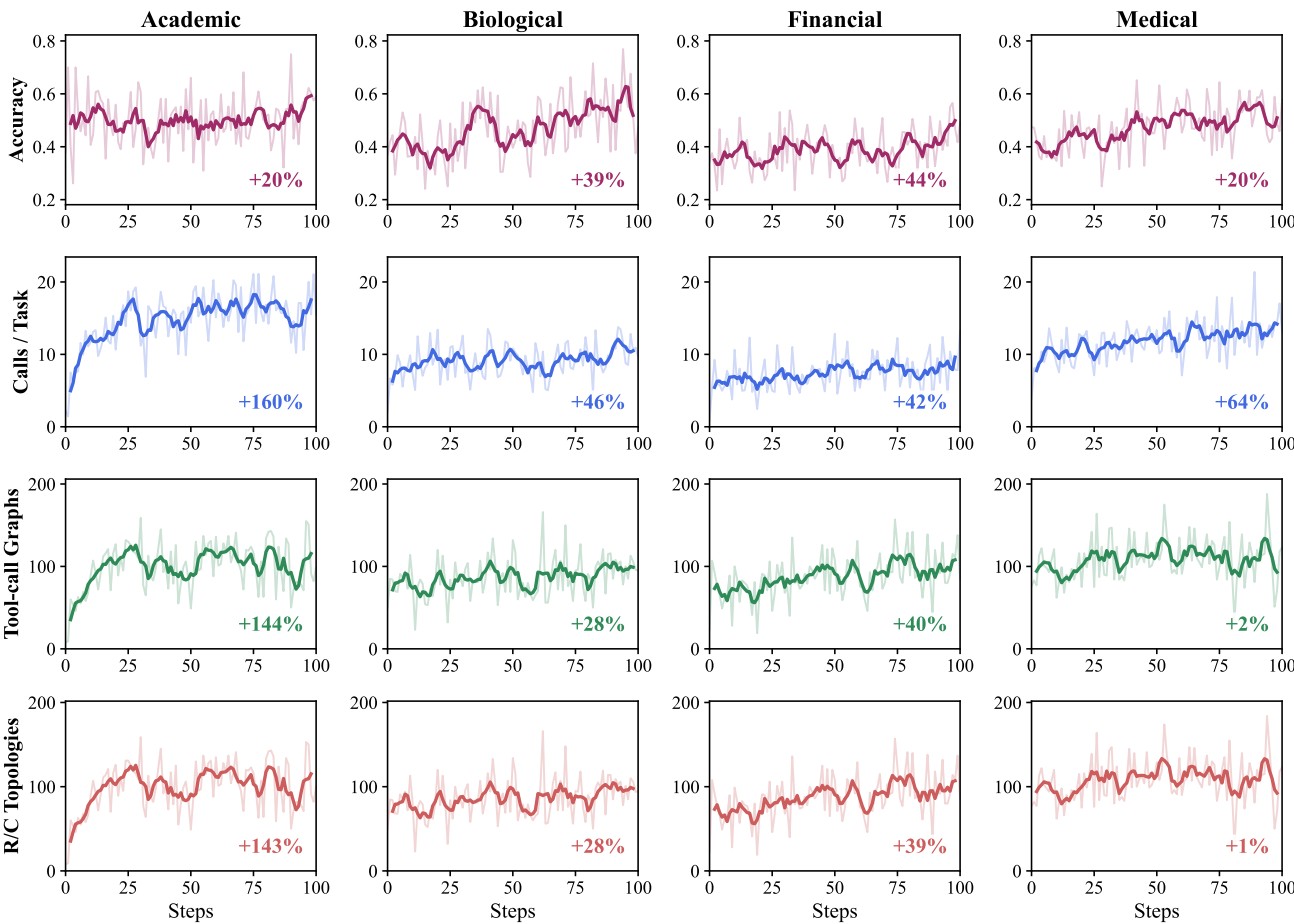

*Figure 6.* **RL training dynamics by domain over 100 steps.** 4 rows (metrics: Accuracy, Calls/Task, Tool-call Graphs, R/P Topologies) × 4 columns (domains: Academic, Biological, Financial, Medical). All domains show consistent accuracy improvement, while structural diversity (tool-call graphs and R/P topologies) also increases during RL. Percentages are relative changes between smoothed start/end values.

## E.1. Scaling Analysis Raw Data

The following tables provide raw experimental data for the scaling analysis in Figure 2.

# F. Training Details

**SFT.** We fine-tune Qwen3-8B in bf16 precision using AdamW ($\beta_1$=0.9, $\beta_2$=0.95, $\epsilon$=1e-8, weight decay 0.1) with a cosine learning rate schedule (warmup 5% of total steps, peak lr 1e-5, min lr 0).

**RL.** We use GRPO with entropy loss enabled, gradient clipping at 1.0, and an off-policy filter (threshold 12.0) to discard stale samples. Rollouts are generated with SGLang (TP=4, group size 8) and a memory fraction of 0.7. Training uses TP=4 and context parallelism (CP=4) with dynamic micro-batch sizing and token-level loss.

*Table 11.* **Raw data for Figure 2(a): Diversity vs. Quantity Scaling.** All results are SFT with 300 steps. **Diversity Scaling**: Fixed 12k trajectories, expanding tool pool (1→4 domains, 174→373 tools). **Quantity Scaling**: Fixed general-purpose search/browse tools (2 tools), expanding data (12k→48k).

| Condition | Setting | $FSC_2$ | $FSC_3$ | FAB | MAB | GAIA | BC | HLE | XB-DS | SWE | Tool. |
|---|---|---|---|---|---|---|---|---|---|---|---|
| *Scaling Diversity (12k fixed, tool pool expanding)* | | | | | | | | | | | |
| 1 domain | fin (174 tools) | 57.1 | 19.0 | 14.0 | 38.8 | 32.0 | 6.5 | 8.5 | 31.2 | 10.9 | 0.9 |
| 2 domain | fin_med (263 tools) | 60.7 | 23.4 | 15.5 | 43.6 | 45.5 | 9.8 | 11.5 | 44.2 | 11.2 | 1.5 |
| 3 domain | fin_med_bio (324 tools) | 61.2 | 27.4 | 17.0 | 45.2 | 47.1 | 10.8 | 12.3 | 48.6 | 11.8 | 2.5 |
| 4 domain | fin_med_bio_aca (373 tools) | 62.0 | 29.8 | 18.0 | 47.6 | 50.5 | 12.2 | 13.4 | 50.1 | 12.5 | 2.9 |
| *Scaling Quantity – Gen-DR (search/browse only, 2 tools fixed)* | | | | | | | | | | | |
| 12k | 1 config | 55.1 | 15.6 | 5.5 | 22.9 | 38.1 | 11.5 | 11.8 | 50.1 | 4.4 | 0.3 |
| 24k | 1 config | 60.1 | 19.0 | 8.5 | 30.9 | 40.9 | 11.1 | 12.4 | 49.2 | 4.3 | 0.9 |
| 36k | 1 config | 57.6 | 17.0 | 6.3 | 33.2 | 42.6 | 11.3 | 12.7 | 50.0 | 4.1 | 0.6 |
| 48k | 1 config | 51.3 | 14.3 | 4.7 | 29.7 | 47.8 | 11.6 | 12.6 | 50.1 | 4.2 | 0.3 |

*Table 12.* **Raw data for Figure 2(b): Config Scaling vs. Pool+Config Scaling.** All results are SFT with 300 steps, starting from Financial domain (12k). **Config Scaling**: Fixed tool pool (fin, 174 tools), expanding data (12k→48k) and configurations. **Pool+Config Scaling**: Jointly expanding tool pool diversity (174→373 tools) and data quantity, all paths starting from fin.

| Condition | Setting | $FSC_2$ | $FSC_3$ | FAB | MAB | GAIA | BC | HLE | XB-DS | SWE | Tool. |
|---|---|---|---|---|---|---|---|---|---|---|---|
| *Config Scaling (fin domain fixed, 174 tools)* | | | | | | | | | | | |
| 12k | fin | 57.1 | 19.0 | 14.0 | 38.8 | 32.0 | 6.5 | 8.5 | 31.2 | 10.9 | 0.9 |
| 24k | fin | 60.6 | 22.9 | 15.0 | 41.3 | 41.6 | 8.9 | 9.8 | 37.6 | 11.3 | 1.2 |
| 36k | fin | 61.6 | 24.2 | 16.0 | 41.8 | 45.5 | 9.8 | 10.5 | 40.5 | 11.5 | 1.5 |
| 48k | fin | 62.1 | 25.3 | 16.5 | 42.0 | 47.0 | 10.3 | 11.0 | 42.8 | 11.5 | 1.5 |
| *Pool+Config Scaling (all paths starting from fin, tool pool expanding)* | | | | | | | | | | | |
| 12k (1d) | fin | 57.1 | 19.0 | 14.0 | 38.8 | 32.0 | 6.5 | 8.5 | 31.2 | 10.9 | 0.9 |
| | fin_med | 60.5 | 25.0 | 16.0 | 48.3 | 50.5 | 8.8 | 10.8 | 48.0 | 11.9 | 3.1 |
| 24k (2d) | fin_bio | 62.2 | 26.2 | 10.0 | 41.8 | 49.5 | 11.6 | 13.6 | 57.0 | 11.5 | 1.4 |
| | fin_aca | 56.3 | 27.4 | 12.0 | 41.9 | 48.5 | 11.1 | 11.0 | 40.0 | 12.4 | 2.5 |
| | fin_med_bio | 60.5 | 30.4 | 24.0 | 50.1 | 47.5 | 11.3 | 14.2 | 47.0 | 12.2 | 2.5 |
| 36k (3d) | fin_med_aca | 62.3 | 27.5 | 22.0 | 49.6 | 49.9 | 11.3 | 12.4 | 54.0 | 13.0 | 3.1 |
| | fin_bio_aca | 60.5 | 26.2 | 18.0 | 43.3 | 52.4 | 12.2 | 12.7 | 46.0 | 13.6 | 4.4 |
| 48k (4d) | fin_med_bio_aca | 62.1 | 33.3 | 27.0 | 50.2 | 50.5 | 13.2 | 13.8 | 50.2 | 13.2 | 4.7 |

*Table 13.* **Scaling diversity & quantity (SFT → RL).** Each cell shows category score after SFT and after RL (SFT→RL). $\Delta$Avg denotes $\text{Avg}_{RL}$-$\text{Avg}_{SFT}$. L2-A averages GAIA/XB-DS/BC/HLE; L2-B averages $\text{FSC}_2$/$\text{FSC}_3$.

| Domains | Combo | L2-A | L2-B | L3-A | L3-B | L3-C | L3-D | Avg | $\Delta$Avg |
|---|---|---|---|---|---|---|---|---|---|
| | fin | 19.6→29.0 | 38.0→44.8 | 14.0→22.0 | 38.8→49.9 | 10.9→11.9 | 0.9→3.7 | **20.4→26.9** | +6.5 |
| | med | 20.8→31.1 | 37.0→44.0 | 12.0→16.0 | 47.1→52.1 | 12.2→12.4 | 2.3→4.4 | 21.9→26.7 | +4.8 |
| 1 domain 12k | aca | 30.1→31.0 | 37.6→46.5 | 10.0→16.0 | 42.3→44.8 | 14.2→14.7 | 1.5→2.5 | 22.6→25.9 | +3.3 |
| | bio | 25.3→30.1 | 38.2→37.1 | 6.0→8.0 | 39.9→50.3 | 11.0→15.9 | 1.0→2.5 | 20.2→24.0 | +3.7 |
| | *Avg* | 23.9→30.3 | 37.7→43.1 | 10.5→15.5 | 42.0→49.3 | 12.1→13.7 | 1.4→3.3 | 21.3→25.9 | +4.6 |
| | fin-med | 29.5→31.8 | 42.8→45.5 | 16.0→26.0 | 48.3→56.4 | 11.9→14.2 | 3.1→4.0 | **25.3→29.7** | +4.4 |
| | fin-bio | 32.9→32.3 | 44.2→45.3 | 10.0→20.0 | 41.8→55.3 | 11.5→13.7 | 1.4→4.4 | 23.6→28.5 | +4.9 |
| | med-bio | 28.1→32.9 | 43.5→43.4 | 14.0→16.0 | 49.8→56.6 | 11.8→12.8 | 2.5→5.3 | 25.0→27.8 | +2.9 |
| 2 domain 24k | aca-med | 28.6→31.0 | 41.0→41.2 | 14.0→16.0 | 46.6→53.4 | 14.2→16.2 | 3.1→5.3 | 24.6→27.2 | +2.6 |
| | aca-fin | 27.6→31.1 | 41.8→47.5 | 12.0→14.0 | 41.9→51.3 | 12.4→13.3 | 2.5→3.7 | 23.1→26.8 | +3.8 |
| | aca-bio | 31.8→34.4 | 43.0→42.6 | 12.0→10.0 | 43.2→52.2 | 13.5→17.3 | 1.4→3.7 | 24.1→26.7 | +2.6 |
| | *Avg* | 29.8→32.2 | 42.7→44.3 | 13.0→17.0 | 45.3→54.2 | 12.6→14.6 | 2.3→4.4 | 24.3→27.8 | +3.5 |
| | fin-med-bio | 30.0→33.6 | 45.5→50.9 | 24.0→32.0 | 50.1→56.6 | 12.2→18.0 | 2.5→6.2 | **27.4→32.9** | +5.5 |
| | fin-bio-aca | 30.8→35.0 | 43.4→49.7 | 18.0→30.0 | 43.3→55.7 | 13.6→17.2 | 4.4→6.5 | 25.6→32.4 | +6.8 |
| 3 domain 36k | fin-med-aca | 31.9→33.6 | 44.9→45.9 | 22.0→28.0 | 49.6→54.8 | 13.0→15.5 | 3.1→4.0 | 27.4→30.3 | +2.9 |
| | med-bio-aca | 29.1→34.5 | 43.9→45.3 | 18.0→18.0 | 48.3→53.9 | 12.6→14.3 | 3.7→5.3 | 25.9→28.5 | +2.6 |
| | *Avg* | 30.5→34.2 | 44.4→48.0 | 20.5→27.0 | 47.8→55.3 | 12.8→16.2 | 3.4→5.5 | 26.6→31.0 | +4.4 |
| 4 domain 48k | aca-fin-med-bio | 31.9→38.4 | 47.7→52.3 | 27.0→34.0 | 50.2→57.3 | 13.2→18.3 | 4.7→8.3 | **29.1→34.8** | +5.6 |
| | *Avg* | 31.9→38.4 | 47.7→52.3 | 27.0→34.0 | 50.2→57.3 | 13.2→18.3 | 4.7→8.3 | 29.1→34.8 | +5.6 |

