# OpenReview forum: "DIVE: Scaling Diversity in Agentic Task Synthesis for Generalizable Tool Use"
_ICML.cc/2026/Conference — ICML 2026 regular_

### Official Review · Reviewer_m9Lh · 2026-03-03

**Soundness:** 3
**Presentation:** 3
**Significance:** 4
**Originality:** 2
**Overall Recommendation:** 5
**Confidence:** 5

**Summary:**

1. Addressing the brittleness of tool-using LLMs caused by insufficient task diversity, the authors propose **DIVE**, a method that reverse-derives tasks from diverse real-world tool executions to ensure robust grounding.
2. DIVE synthesizes 48k trajectories across 374 tools, generating nearly 40,000 unique toolsets and tool-call graphs to maximize structural diversity.
3. Training Qwen3-8B on this data improves performance by **+22 points** on out-of-distribution benchmarks, demonstrating that scaling diversity consistently outperforms scaling data quantity.

**Compliance With Llm Reviewing Policy:**

Affirmed.

**Final Justification:**

Most of the problems are solved and I will raise the accept score

**Key Questions For Authors:**

**Questions for Authors:**

1. **Data Release Timeline**: Could you please clarify when you plan to open-source the constructed dataset? Making this available would significantly benefit the community and enable reproducibility.

2. **SFT vs. RL Data Analysis**: Could you provide a more detailed analysis of the SFT and RL data distribution and their respective contributions to the final performance? From Table 2, it appears that most improvements come from SFT training rather than RL. A breakdown of the performance gains from each training stage would be helpful.

3. **Verification Mechanism in RL Training**: The authors claim the tasks are "verifiable," but the verification method used during RL training is not clearly specified. Is verification rule-based (checking tool-call correctness) or LLM-based (evaluating final answers)? While Appendix B.3 suggests an LLM-based checker, if this is the case, I would argue that these are not truly verifiable tasks. The potential biases and limitations of LLM-based evaluators should be explicitly acknowledged and discussed in detail.

4. **Task Complexity**: Are the tasks designed for single-step or multi-step tool use? Clarifying the average number of tool calls per task would help readers better understand the complexity of the benchmark.

5. **End-to-End RL Training Support**: Does the framework support end-to-end RL training with real-time tool execution? Since the toolset is API-based, it appears that real-time tool call responses during training may not be feasible. Could the authors clarify how tool interactions are handled during the RL training phase?

**Limitations:**

No.

The authors are encouraged to include a dedicated 'Limitations' section. Explicitly acknowledging the paper's weaknesses is highly valuable, even if they are not fully resolved in this work.

**Strengths And Weaknesses:**

## Strengths
- This is a high-quality and significant paper that makes a valuable contribution to the community.
- The authors constructed a pipeline to synthesize diverse tool-use data, which aligns perfectly with current community needs.
- They successfully developed 374 robust tools grounded in real-world scenarios.
- The experimental evaluation is comprehensive, and the overall methodology is sound.

## Weaknesses
See Key Questions For Authors.

---

> ### Author Rebuttal · Authors · 2026-03-30
>
> We thank the reviewer for the positive assessment and constructive suggestions. We address each question below.
>
> > **Q1: Could you clarify when you plan to open-source the constructed dataset?**
>
> We will release the full dataset, trained model weights, and tool pool upon acceptance. Code is already prepared for release.
>
> > **Q2: Could you provide a more detailed analysis of the SFT and RL data distribution and their respective contributions?**
>
> SFT serves as a cold start that establishes reliable tool-calling behavior, lifting the average by +17.6 points across 9 OOD benchmarks (Table 2). RL adds +5.9 on average, with strong gains on challenging benchmarks (GAIA +11.9, FAB +6.0, SWE +5.1). While smaller in magnitude, we argue RL plays a qualitatively different role: it **accelerates the diversity-generalization scaling trend**.
>
> We summarize the marginal gains per diversity step from the 24-path scaling experiment (Table 5, Fig. 3c):
>
> | Diversity step | SFT marginal gain | RL marginal gain |
> |---|---|---|
> | 1→2 domains | +3.0 | +1.9 |
> | 2→3 domains | +2.3 | **+3.2** |
> | 3→4 domains | +2.5 | **+3.8** |
> | **Total (1→4)** | **+7.8** | **+8.9** |
>
> SFT's marginal gains are flat or slightly diminishing as diversity grows (3.0→2.3→2.5), suggesting saturation of what imitation alone can extract. In contrast, RL's marginal gains **accelerate** (1.9→3.2→3.8): the more diverse the training data, the more RL benefits. This suggests a complementary relationship: SFT provides a strong foundation, but RL is uniquely positioned to exploit the exploration landscape that diverse data opens up. Our structural diversity analysis (§5.2, Fig. 4) provides a mechanistic explanation: during RL, unique tool-call graphs and R/P topologies continue to grow over 100 steps, confirming the model discovers new effective tool-use structures beyond SFT demonstrations.
>
> > **Q3: Is verification rule-based or LLM-based? The potential biases and limitations of LLM-based evaluators should be discussed.**
>
> Thank you for raising this important point. Our verification is LLM-based rather than strict rule matching. We chose this design because every synthesized task has a deterministic reference answer derived directly from tool-returned outputs, which is typically a concise factual string (e.g., protein IDs like "Q6A0I4", stock codes like "603658.SH"; see Appendix D). The verification task is therefore comparing two short answers rather than evaluating open-ended reasoning or lengthy trajectories. For RL specifically, this is preferable to rigid rule matching: it tolerates minor formatting differences while correctly rejecting wrong answers, which provides cleaner reward signals.
>
> To ensure accuracy, we employ dual-model cross-verification where Claude-4-Sonnet and DeepSeek-V3.2 must independently agree (Appendix B.3). A 200-sample human audit confirmed 100% agreement. We acknowledge that LLM judges may exhibit leniency on partially correct answers or have blind spots on uncommon domain formats. The concise factual nature of our answers and the dual-model design mitigate but do not fully eliminate these risks. This form of ground truth and LLM-based evaluation is consistent with mainstream agentic benchmarks (GAIA, HLE, BrowseComp).
>
> > **Q4: Are the tasks single-step or multi-step? What is the average number of tool calls per task?**
>
> Our tasks are multi-step. From the 48k SFT trajectories (Table 4), the average task involves **11.1 tool calls**, ranging from simple 2-3 call tasks to complex 40+ call tasks. During RL training, tool calls per task further increase by ~35% as the model learns to explore more thoroughly (Fig. 4).
>
> > **Q5: Does the framework support end-to-end RL training with real-time tool execution?**
>
> Yes. During RL training, the student policy interacts with live API-based tools: at each step, tool calls are executed against real endpoints and observations are returned. To ensure training stability, every tool passed rigorous validation before entering the pool (§3.1):
>
> | Metric | Description | Result |
> |---|---|---|
> | Basic functionality | Successful execution with valid inputs | 100% |
> | Avg concurrency capacity | Max concurrent requests sustained (≥85% success) | 869 |
> | Result consistency | Same output for identical inputs (10 repeated calls) | 100% |
> | Concurrent stability | Sustained load test (100-conc. × 3 rounds, ≥80% success) | 100% |
> | Avg response time | Mean single-call latency | 3.48s |
>
> Tools failing any test were discarded.
>
> Additionally, following Reviewer V2zV's suggestion, we evaluated on two new benchmark suites: BFCL v3 [1] and τ-bench/τ³-bench [2] (7 sub-tasks total), where DIVE improves over the base by **+15.6 points (+65%)** on average. Full results and 5 new baselines are reported in our response to Reviewer V2zV.
>
> [1] Yan et al. Berkeley Function Calling Leaderboard v3. 2024.
>
> [2] Yao et al. τ-bench: A Benchmark for Tool-Agent-User Interaction. 2024.

---

> > ### Author Rebuttal · Reviewer_m9Lh · 2026-04-03
> >
> > Q4: Support for Multi-turn Tool Invocation:
> > Could the authors clarify whether and how the proposed framework supports multi-turn tool interactions that involve iterative tool execution and environment feedback?
> >
> > Q5: End-to-End RL Training Support, really live API-based tools?
> > Could the authors provide concrete examples or a taxonomy of the supported tools.
> > It is not clarify how tool interactions are handled during the RL training phase.
> > Furthermore, it is unclear how tool interactions are handled during the RL training phase—specifically, whether the environment simulates tool responses or interacts with live endpoints, and how latency or non-determinism is managed.

---

> > > ### Author Response · Authors · 2026-04-03
> > >
> > > Thank you for the follow-up. We clarify both points below.
> > >
> > > > **Q4: Whether and how the framework supports multi-turn tool interactions with iterative execution and environment feedback.**
> > >
> > > Yes. Our RL infrastructure decouples the agent, tool-serving middleware (gateway MCP server), and training engine into separate modules, enabling scalable multi-turn agentic training. During RL, the agent performs multi-turn tool interactions:
> > >
> > > **Within a rollout (multi-turn):** The agent generates reasoning (`reasoning_content`), then issues a tool call. All API tools are wrapped as **MCP servers behind a unified gateway**, which routes each call to the corresponding real endpoint and returns the **real execution response** as a `tool_response` message appended to the conversation context. The agent reasons over the updated context and may issue further tool calls. This **iterative loop** continues until the agent produces a final answer without any tool call.
> > >
> > > **After a rollout (verification):** The final answer is compared against the reference answer via dual-model cross-verification (Appendix B.3) to produce the reward signal for GRPO policy updates.
> > >
> > > **Across rollouts (parallelism):** Rollouts execute asynchronously using a **windowed FIFO scheduling strategy** that balances throughput and training stability. Since each task involves different domains and toolsets, the concurrent API load is naturally distributed across endpoints. All live APIs passed rigorous validation for consistency, concurrency, and latency before entering the pool (§3.1).
> > >
> > > A simplified example from our SFT data:
> > >
> > > ```json
> > > {"messages": [
> > >   {"role": "system", "content": "You are a helpful assistant.\n[tool definitions injected]"},
> > >   {"role": "user", "content": "Among papers published in 2025 ..."},
> > >   {"role": "assistant", "content": "", "reasoning_content": "I need to search PubMed ...",
> > >    "tool_calls": [{"function": {"name": "ncbi_entrez_search", ...}}]},
> > >   {"role": "tool", "tool_name": "ncbi_entrez_search",
> > >    "content": "{\"Count\": \"178\", \"IdList\": [\"40985725\", ...]}"},
> > >   {"role": "assistant", "content": "", "reasoning_content": "178 results. Let me refine ...",
> > >    "tool_calls": [{"function": {"name": "ncbi_entrez_summary", ...}}]},
> > >   {"role": "tool", "tool_name": "ncbi_entrez_summary",
> > >    "content": "{\"records\": [{\"PubDate\": \"2025 Sep 12\", ...}]}"},
> > >   // ... 8 tool calls total, each with reasoning + real API response ...
> > >   {"role": "assistant", "content": "PMID: 40903045, Journal: Assay and Drug Development Technologies ..."}
> > > ], "tools": [30 available tools], "metadata": {"domain": "biological"}}
> > > ```
> > >
> > > > **Q5: Concrete tool examples, live vs. simulated execution, and latency/non-determinism management.**
> > >
> > > **Tool taxonomy.** The paper provides a **complete tool listing in the appendix**: Table 6 (Retrieval vs. Processing distribution across 5 domains), Table 7 (all API sources with documentation links), and Table 8 (complete list of all 374 tools). All tools are wrapped as **MCP servers routing to real API endpoints** (no cached or simulated responses). Representative examples:
> > >
> > > | Domain | Source | Example tools |
> > > |---|---|---|
> > > | Financial | Tushare | `stock_hsgt_top10`, `fund_portfolio` |
> > > | Medical | RxNorm, RxClass... | `rxnorm_find_rxcui_by_string`, `rxclass_get_class_members` |
> > > | Biological | NCBI Entrez, BioPython... | `ncbi_entrez_search`, `bio_seq_translate` |
> > > | Academic | Crossref, OpenAlex, arXiv... | `crossref_get_work`, `openalex_search_authors` |
> > > | General | Google, Python sandbox... | `search`, `jupyter_execute_code_cell` |
> > >
> > > **Non-determinism.** We designed a dedicated consistency test to filter live public APIs: each tool must return **identical results across 10 repeated calls** with the same input (100% pass required, §3.1). Tools exhibiting non-deterministic behavior are excluded. Furthermore, tools involving time-sensitive data require explicit temporal parameters (e.g., `stock_hsgt_top10(trade_date="20260402")`), so queries are anchored to specific dates rather than returning "latest" results.
> > >
> > > **Latency.** We also filter out any public API tool with **concurrency capacity** below 100 or **response time** exceeding 30s. The retained tools average 3.48s response time and sustain 869 concurrent requests. Processing tools (code execution) run locally with near-zero latency. Combined with the **windowed FIFO scheduling and distributed API load** described above, this keeps RL training stable and efficient.
> > >
> > > We hope this fully addresses the reviewer's concerns. We will expand the RL training details in the revision.

---

### Official Review · Reviewer_iRLq · 2026-03-08

**Soundness:** 2
**Presentation:** 3
**Significance:** 2
**Originality:** 2
**Overall Recommendation:** 3
**Confidence:** 3

**Summary:**

This paper presents DIVE, a framework designed to improve the generalization of LLM-based agents in tool-use tasks. The authors identify a "brittleness" in current agents caused by training on synthesized data that lacks structural and toolset diversity. To solve this, DIVE inverts the typical synthesis pipeline: instead of generating a query first, it executes real-world tools to collect "evidence" (execution traces) and then reverse-derives grounded tasks that are strictly supported by those traces. The framework utilizes a pool of 374 real-world tools across five domains and employs an iterative synthesis loop to increase task complexity. Experimental results on Qwen3-8B show that DIVE significantly outperforms specialized baselines on out-of-distribution benchmarks, demonstrating that scaling tool diversity is more effective than scaling data quantity alone

**Compliance With Llm Reviewing Policy:**

Affirmed.

**Key Questions For Authors:**

Please refer to weakness.

**Limitations:**

yes

**Strengths And Weaknesses:**

**Strength**
1. The paper is well-structured, with clear visualizations of the framework and the scaling trends. The inclusion of detailed case studies in the appendix helps illustrate the complexity of the synthesized tasks
2. The work offers a perspective shift from scaling task-content diversity to scaling toolset-interaction diversity. It argues that the bottleneck for generalization is exposure to diverse tool-use "topologies" rather than just varied query phrasings.


**Weakness**
1. A potential flaw lies in the requirement for the "Evidence Collector" to call tools somewhat arbitrarily to build the initial trace. The validity of the "Reasoning" steps generated during task derivation is questionable. If the task generator LLM produces reasoning that merely justifies the connection between evidence and the random tool-calls, it might not provide a high-quality "thought" chain for the student model to learn actual problem-solving logic. This could lead to a model that is good at matching patterns between tools but lacks robust, first-principles reasoning
2. The process of reverse-deriving a coherent, natural language task from a "cloud" of evidence is non-trivial. There is a risk that the resulting questions are "tailored" too closely to the specific tool outputs found, leading to queries that feel artificial or "reverse-engineered" rather than representing authentic user intent.
3. The framework involves tools across multiple specialized domains, which suggests that a comparison with domain-specific tool agents (e.g., specialized agents for Finance or Medicine) is necessary to fully validate its effectiveness. Currently, the evaluation only includes a comparison with a software engineering agent SWE-Dev-8B which may be considered an outdated baseline for that specific domain.

---

> ### Author Rebuttal · Authors · 2026-03-30
>
> We thank the reviewer for the feedback.
>
> > **W1: The Evidence Collector calls tools arbitrarily... this could lead to a model that matches patterns but lacks first-principles reasoning.**
>
> We clarify a key distinction in our pipeline (Figure 2):
>
> - **Synthesis stage** (§3.2): The collector gathers grounded evidence for task derivation, not to demonstrate solution steps. Starting from a seed, it expands related tool-call results. The Task Generator derives task tuples $(Q, A, \mathcal{T})$ from this evidence. The generator outputs only query-answer pairs; neither the collector's exploration traces nor any intermediate reasoning enter the training data.
> - **Training stage** (§3.3): A teacher solves each derived task from scratch via ReAct reasoning, without access to collector traces. Only verified successful trajectories are retained for SFT. For RL, the student explores independently with reward feedback.
>
> Evidence collection is not arbitrary: the collector reasons over prior results at each step, constrained by seed concepts, domain-conditioned toolsets (15-50 tools), exemplars, and an iterative loop. Tasks average 11.1 tool calls (Table 4). If the model had learned only shallow patterns, we would expect degradation under tool shift. The opposite holds: DIVE shows its largest gains on the most OOD benchmarks (Toolathlon 0.9→8.3, SWE 10.8→18.3). Our scaling analysis (§5.1) confirms diversity scaling outperforms quantity scaling.
>
> > **W2: The resulting questions are "tailored" too closely to specific tool outputs, leading to queries that feel artificial rather than representing authentic user intent.**
>
> The reviewer raises a valid concern. Our primary objective is structural diversity of tool-use patterns (§5.1), but this does not sacrifice query quality. The Task Generator is conditioned on **exemplars** drawn from real task families (§3.1), which provide authentic query structures as priors. Derived queries thus reflect natural task forms (e.g., comparative analysis, multi-hop lookup) while their content is grounded in real tool outputs. Moreover, all evidence is collected around a single seed topic through step-by-step LLM reasoning, so the information available for task derivation is thematically unified and internally coherent, rather than assembled from unrelated fragments. The OOD results (+22 avg points on 9 independently developed benchmarks) confirm this approach effectively trains generalizable skills.
>
> > **W3: A comparison with domain-specific tool agents is necessary to fully validate its effectiveness.**
>
> We add two domain-specific agents (**II-Medical-8B** [4], **Fino1-8B** [5]) and three general tool-use agents (**xLAM-2-8B** [1], **ToolACE-8B** [2], **TOUCAN-7B** [3]), alongside existing baselines:
>
> | Model | GAIA | HLE | BC | XB-DS | FSC₂ | FSC₃ | FAB | MAB | SWE | Tool. |
> |---|---|---|---|---|---|---|---|---|---|---|
> | xLAM-2-8B | 8.9 | 2.3 | 1.1 | 30.0 | 37.8 | 1.2 | 2.0 | 0.0 | 1.2 | 0.3 |
> | ToolACE-8B | 9.7 | 2.2 | 0.3 | 9.0 | 9.3 | 1.2 | 2.0 | 2.3 | 0.0 | 0.0 |
> | TOUCAN-7B | 10.7 | 5.5 | 0.9 | 17.0 | 31.9 | 1.2 | 6.0 | 15.4 | 2.6 | 0.3 |
> | II-Medical-8B | 20.6 | 6.6 | 1.7 | 37.0 | 35.3 | 3.6 | 8.0 | 41.1 | 3.3 | 0.3 |
> | Fino1-8B | 5.8 | 4.2 | 0.4 | 9.0 | 16.9 | 1.2 | 2.0 | 5.5 | 0.0 | 0.0 |
> | EnvScaler-8B | 25.8 | 2.8 | 1.7 | 45.7 | 40.7 | 10.8 | 14.0 | 56.6 | 11.5 | 2.2 |
> | WebExplorer-8B | 50.0 | 17.3 | 15.7 | 53.7 | 35.9 | 18.1 | 4.0 | 17.8 | 7.0 | 0.3 |
> | Qwen3-8B (base) | 22.4 | 6.4 | 1.3 | 24.0 | 28.6 | 7.1 | 2.0 | 38.4 | 10.8 | 0.9 |
> | **DIVE-8B (RL)** | **61.2** | **17.8** | **16.4** | **58.1** | **67.3** | **37.3** | **34.0** | **57.3** | **18.3** | **8.3** |
>
> Domain specialists show strength in related domains (II-Medical: 41.1 on MAB) but transfer poorly elsewhere. DIVE outperforms II-Medical on MAB (57.3 vs. 41.1) without medical data. We also evaluated on BFCL v3 [6] and τ-bench/τ³-bench [7]:
>
> | Model            | BFCL-base | BFCL-mp  | BFCL-lc  | BFCL-search | τ-retail | τ-airline | τ³-banking |
> | ---------------- | --------- | -------- | -------- | ----------- | -------- | --------- | ---------- |
> | Qwen3-8B (base)  | 32.0      | 22.0     | 28.0     | 17.0        | 38.2     | 28.0      | 2.1        |
> | **DIVE-8B (RL)** | **48.0**  | **42.0** | **38.5** | **61.0**    | **49.1** | **30.0**  | **8.2**    |
>
> DIVE improves over the base by **+15.6 points (+65%)** on average across these 7 sub-tasks.
>
> **References:**
> [1] Zhang et al. xLAM-2: Large Action Models for Multi-Turn Agent Tasks. Salesforce, 2025.
>
> [2] Liu et al. ToolACE: Winning the Points of LLM Function Calling. ICLR 2025.
>
> [3] Gao et al. TOUCAN: Tool-Use through Chain-of-Thought and Action. 2025.
>
> [4] Intelligent-Internet. II-Medical-8B. HuggingFace, 2025.
>
> [5] Qian et al. Fino1: On the Transferability of Reasoning-Enhanced LLMs to Finance. NeurIPS 2025.
>
> [6] Yan et al. Berkeley Function Calling Leaderboard v3. 2024.
>
> [7] Yao et al. τ-bench: A Benchmark for Tool-Agent-User Interaction. 2024.

---

> > ### Author Rebuttal · Reviewer_iRLq · 2026-04-05
> >
> > The authors’ clarification about using a single seed topic per evidence collection trajectory helps explain why the resulting tasks are thematically coherent and less likely to be incoherent “Frankenstein” compositions. This design indeed addresses part of my concern about unnatural reverse-engineered questions. At the same time, this raises a natural follow-up question about coverage: if each synthesized task cluster is anchored to a specific seed, then the overall diversity of tool-use structures and domains depends critically on the quality and quantity of these seeds. I encourage the authors to explicitly describe how seeds are selected and distributed across domains and topics, and how this seed design ensures both the diversity and the quality of the synthesized tasks.

---

> > > ### Author Response · Authors · 2026-04-05
> > >
> > > We thank the reviewer for the continued engagement and for acknowledging that the seed-topic coherence design addresses part of the concern. This is an important question. Seed design is central to how DIVE achieves diversity, and we are happy to elaborate.
> > >
> > > > **Follow-up: How are seeds selected and distributed? How does this ensure both diversity and quality?**
> > >
> > > **Seed selection and distribution.** As described in §3.1, we mine **~5,000 entity seeds per domain** across 5 domains from authoritative large-scale sources: Wikipedia, PubMed, NCBI, and global stock exchanges. These sources naturally provide broad coverage of sub-topics within each domain (e.g., medical seeds span oncology, cardiology, pharmacology, etc.), ensuring long-tail semantic coverage rather than clustering in a few popular areas. Seeds are deliberately specific entities (e.g., "Erlotinib", "Q6A0I4", "603658.SH") rather than generic categories (e.g., "medicine"), which anchors each synthesis to a concrete, information-rich topic and avoids topic collapse.
> > >
> > > **Seeds are not the sole driver of diversity.** A key design principle of DIVE is that diversity arises from the **combinatorial composition** of three decoupled resources: seeds, toolsets, and exemplars (§3.1). Each synthesis cycle independently samples a (seed, toolset, exemplar set) configuration. The same seed can be sampled multiple times, and each time paired with different toolsets (15–50 tools randomly sampled) and different exemplar priors, producing structurally different tasks. This combinatorial mechanism is what scales diversity beyond what any single axis (including seeds) could provide alone.
> > >
> > > The quantitative evidence confirms this: 48k trajectories yield **46,398 unique toolsets** and **39,810 unique tool-call graphs**, far exceeding the number of seeds, demonstrating that the combinatorial design is effective.
> > >
> > > **Quality.** After mining raw entities from data sources, we apply LLM-based deduplication and filtering to remove near-duplicates and low-information seeds. Each retained seed is a real-world entity with verifiable information accessible through our tool pool, ensuring the evidence collector retrieves substantive, factual data. The iterative loop ($K$=3) then progressively deepens evidence and increases task complexity (Appendix C), and rejection sampling retains only verified successful trajectories.
> > >
> > > We will make the seed selection strategy and its role in diversity more explicit in the revision.

---

### Official Review · Reviewer_V2zV · 2026-03-13

**Soundness:** 3
**Presentation:** 3
**Significance:** 3
**Originality:** 3
**Overall Recommendation:** 5
**Confidence:** 3

**Summary:**

This paper proposes DIVE, a method to synthetically curate more data for agentic task synthesis while ensuring diversity. They do this by inverting the traditional synthesis order - rather than generating tasks and then traces, they generate traces and then derive the task from them. More broadly, they also examine diversity data requirements for tool usage - grounded validty and structural validity. Using the dataset generated by DIVE, the tuned models improved across domains, especially compare to other models and baselines.

**Compliance With Llm Reviewing Policy:**

Affirmed.

**Final Justification:**

My two concerns were missing 8B baselines (xLAM-2-8B, ToolACE-8B, TOUCAN) and missing standard benchmarks (BFCL, τ-Bench). The rebuttal added both and the method performs well on them. The diversity-over-quantity scaling result is the paper's strongest contribution and is now supported by a more complete evaluation. Raising to 5 (accept).

**Key Questions For Authors:**

None

**Limitations:**

Not sufficiently, they should talk more about the societal impacts of this work and limitations

**Strengths And Weaknesses:**

Strengths
* The paper is well-written and motivates the problem nicely. The figures are nicely done and easy to read.
* The proposed method is well-explained and intuitive. It is a nice take on traditional synthetic data generation.
* The experiments are well done and the proposed method works well. Authors put lots of effort into doing both SFT and RL, as well as taxonomizing their evaluation (L1/L2/L3) and running several benchmarks and other models, including similar size and other frontier models. This is convincing to show that the proposed DIVE methodology is helpful.
* The scaling analysis graphs are thorough and insightful, showing how much better diversity-only scaling performs better than the quality-only counterpart

Weaknesses
* Comparison with several other 8B models are sparse - it would be helpful to add xLAM-2-8B (Salesforce, April 2025), ToolACE-8B (ICLR 2025), and TOUCAN (https://arxiv.org/pdf/2510.01179)
* Several popular tool-calling benchmarks are omitted - BFCL and TauBench. It would be more convincing to include these.

---

> ### Author Rebuttal · Authors · 2026-03-30
>
> We thank the reviewer for the positive assessment and constructive suggestions. We have conducted extensive additional experiments, adding 5 new 8B baselines and 2 new benchmark suites (7 sub-tasks total).
>
> > **W1: Comparison with several other 8B models are sparse - it would be helpful to add xLAM-2-8B, ToolACE-8B, and TOUCAN.**
>
> We agree that broader baseline coverage strengthens the evaluation. We add all three requested general tool-use agents (**xLAM-2-8B** [1], **ToolACE-8B** [2], **TOUCAN-7B** [3]) and two domain-specific agents (**II-Medical-8B** [4] for medicine, **Fino1-8B** [5] for finance), evaluated on all L2/L3 benchmarks under the same protocol:
>
> | Model            | GAIA     | HLE      | BC       | XB-DS    | FSC₂     | FSC₃     | FAB      | MAB      | SWE  | Tool.   |
> | ---------------- | -------- | -------- | -------- | -------- | -------- | -------- | -------- | -------- | ---- | ------- |
> | xLAM-2-8B        | 8.9      | 2.3      | 1.1      | 30.0     | 37.8     | 1.2      | 2.0      | 0.0      | 1.2  | 0.3     |
> | ToolACE-8B       | 9.7      | 2.2      | 0.3      | 9.0      | 9.3      | 1.2      | 2.0      | 2.3      | 0.0  | 0.0     |
> | TOUCAN-7B        | 10.7     | 5.5      | 0.9      | 17.0     | 31.9     | 1.2      | 6.0      | 15.4     | 2.6  | 0.3     |
> | II-Medical-8B    | 20.6     | 6.6      | 1.7      | 37.0     | 35.3     | 3.6      | 8.0      | 41.1     | 3.3  | 0.3     |
> | Fino1-8B         | 5.8      | 4.2      | 0.4      | 9.0      | 16.9     | 1.2      | 2.0      | 5.5      | 0.0  | 0.0     |
> | EnvScaler-8B     | 25.8     | 2.8      | 1.7      | 45.7     | 40.7     | 10.8     | 14.0     | 56.6     | 11.5 | 2.2     |
> | WebExplorer-8B   | 50.0     | 17.3     | 15.7     | 53.7     | 35.9     | 18.1     | 4.0      | 17.8     | 7.0  | 0.3     |
> | Qwen3-8B (base)  | 22.4     | 6.4      | 1.3      | 24.0     | 28.6     | 7.1      | 2.0      | 38.4     | 10.8 | 0.9     |
> | **DIVE-8B (RL)** | **61.2** | **17.8** | **16.4** | **58.1** | **67.3** | **37.3** | **34.0** | **57.3** | **18.3** | **8.3** |
>
> The expanded comparison reinforces our core claim. General tool-use agents (xLAM, ToolACE, TOUCAN) struggle across OOD benchmarks, further underscoring the challenge of OOD generalization in tool use. Domain-specific models show strength in their related domain (e.g., II-Medical-8B reaches 41.1 on MAB) but transfer poorly elsewhere (Toolathlon 0.3, SWE 0.0). Notably, DIVE outperforms II-Medical even on MAB (57.3 vs. 41.1) despite having no medical-domain data. This is consistent with our thesis that diversity-driven synthesis produces robust generalization.
>
>
> > **W2: Several popular tool-calling benchmarks are omitted - BFCL and TauBench. It would be more convincing to include these.**
>
> Following the reviewer's suggestion, we evaluate all models on BFCL v3 [6] (multi-turn: base, missing-parameter, long-context; agentic web-search) and τ-/τ³-bench [7] (retail, airline, banking):
>
> | Model | BFCL-base | BFCL-mp | BFCL-lc | BFCL-search | τ-retail | τ-airline | τ³-banking |
> |---|---|---|---|---|---|---|---|
> | xLAM-2-8B | 31.5 | 44.2 | 34.5 | 2.0 | 51.1 | 36.0 | 3.3 |
> | ToolACE-8B | 2.0 | 1.0 | 1.5 | 0.0 | 7.8 | 18.0 | 3.1 |
> | TOUCAN-7B | 28.0 | 15.0 | 15.5 | 3.0 | 27.8 | 6.0 | 1.1 |
> | II-Medical-8B | 8.5 | 5.0 | 6.5 | 14.0 | 7.0 | 0.0 | 4.0 |
> | Fino1-8B | 3.0 | 1.5 | 2.0 | 4.0 | 7.0 | 32.0 | 0.0 |
> | EnvScaler-8B | 55.5 | 35.0 | 41.0 | 25.0 | 53.6 | 34.0 | 7.2 |
> | WebExplorer-8B | 30.5 | 17.5 | 17.5 | 46.0 | 7.8 | 24.0 | 7.1 |
> | Qwen3-8B (base) | 32.0 | 22.0 | 28.0 | 17.0 | 38.2 | 28.0 | 2.1 |
> | **DIVE-8B (RL)** | **48.0** | **42.0** | **38.5** | **61.0** | **49.1** | **30.0** | **8.2** |
>
> The same pattern holds: each baseline shows high variance across settings. xLAM performs well on τ-retail (51.1) with structured function calls but collapses on BFCL-search (2.0) and τ³-banking (3.3) which require agentic web interaction and knowledge-intensive reasoning. WebExplorer reaches 46.0 on BFCL-search but drops to 7.8 on τ-retail. DIVE remains consistently competitive across all 7 sub-tasks, improving over the base model by **+15.6 points on average (+65%)**. This aligns with our central thesis: scaling diversity produces balanced generalization rather than narrow peaks.
>
> **References:**
> [1] Zhang et al. xLAM-2: Large Action Models for Multi-Turn Agent Tasks. Salesforce, 2025.
>
> [2] Liu et al. ToolACE: Winning the Points of LLM Function Calling. ICLR 2025.
>
> [3] Gao et al. TOUCAN: Tool-Use through Chain-of-Thought and Action. 2025.
>
> [4] Intelligent-Internet. II-Medical-8B. HuggingFace, 2025.
>
> [5] Qian et al. Fino1: On the Transferability of Reasoning-Enhanced LLMs to Finance. NeurIPS 2025.
>
> [6] Yan et al. Berkeley Function Calling Leaderboard v3. 2024.
>
> [7] Yao et al. τ-bench: A Benchmark for Tool-Agent-User Interaction. 2024.

---

> > ### Author Rebuttal · Reviewer_V2zV · 2026-04-05
> >
> > Thank you for your rebuttal, which adds significant new experiments, demonstrating strong empirical results for this method compared to the additional baselines and on the additional datasets. My concerns are mostly addressed and I have adjusted my score(s) accordingly.

---

> > > ### Author Response · Authors · 2026-04-07
> > >
> > > We sincerely thank the reviewer for the thoughtful engagement and for raising the score. We are very grateful for the recognition of our additional experiments, and we will incorporate the new baselines and benchmarks into the revised version.

---

### Decision · Program_Chairs · 2026-04-30

**Decision:**

Accept (regular)

**Comment:**

The paper proposes DIVE, an evidence-driven framework designed to improve the generalization of tool-using large language models (LLMs). To address the brittleness caused by low-diversity training data, DIVE inverts the traditional task synthesis pipeline: it executes real-world tools first to collect "evidence" traces, and then reverse-derives verifiable and executable queries anchored to those traces. The framework leverages 374 real-world tools across five domains. All three reviewers recognized the importance of the problem and praised the paper's methodology and thorough evaluation.

Strengths:
- Intuitive and Novel Approach: Reviewer V2zV found the inverted synthesis method intuitive and a refreshing take on synthetic data generation. Reviewer iRLq appreciated the perspective shift from scaling task-content diversity to scaling toolset-interaction diversity.
- Comprehensive Methodology: Reviewer m9Lh highlighted the construction of a pipeline with 374 robust, real-world tools, noting it aligns perfectly with current community needs.
- Strong Empirical Results: The evaluation was highly praised. Reviewer V2zV commended the thorough scaling analysis, which convincingly demonstrated that scaling diversity outperforms merely scaling data quantity. Reviewer m9Lh also praised the comprehensive experimental evaluation and sound methodology.

Weaknesses and Rebuttal Discussions:
-  Missing Baselines and Benchmarks: Reviewer V2zV pointed out the absence of several 8B baselines (e.g., xLAM-2-8B, ToolACE-8B) and standard benchmarks like BFCL and TauBench. Reviewer iRLq similarly requested comparisons with domain-specific tool agents. In response, the authors executed a massive evaluation, adding 5 new baselines and 2 new benchmark suites (7 sub-tasks total). This new data reinforced their core claims, fully resolving Reviewer V2zV's concerns.
- Reviewer iRLq expressed concern that the "Evidence Collector" might call tools arbitrarily, leading to reverse-engineered or artificial queries lacking first-principles reasoning. The authors clarified that queries are anchored by natural exemplars and specific seed topics, maintaining thematic coherence rather than assembling unrelated fragments. In a follow-up, the authors detailed their process of mining ~5,000 specific entity seeds per domain from authoritative sources, satisfying the reviewer's questions about diversity and quality coverage.
- RL Verification and Live API Execution: Reviewer m9Lh questioned the reliability of LLM-based verification during RL and asked for clarification on how the framework handles end-to-end real-time tool execution. The authors clarified that they use a dual-model cross-verification system (which achieved 100% agreement in human audits) because the reference answers are concise and factual. Furthermore, they explained their MCP server architecture for routing live API calls, backed by rigorous consistency tests and latency filters, which effectively addressed the reviewer's operational concerns.

The initial scores and reviews reflected a strong baseline appreciation for the paper, combined with requests for a more robust baseline comparison. Reviewer V2zV raised their score to a 5 (Accept) following the inclusion of the requested benchmarks. Reviewer m9Lh, after an extended technical discussion regarding the RL environment and live API infrastructure, also confidently recommended an Accept (5). While Reviewer iRLq leaned towards a Weak Reject (3) due to concerns regarding task artificiality , the authors' detailed breakdown of seed coherence and combinatorial scaling mitigated these concerns. The submission is technically solid, features an exceptionally strong rebuttal, and makes a valuable contribution to agentic LLM training.